# METTL3-mediated chromatin contacts promote stress granule phase separation through metabolic reprogramming during senescence

Chen Wang[1], Hideki Tanizawa[2,3], Connor Hill[4], Aaron Havas[5], Qiang Zhang[4], Liping Liao[1], Xue Hao [1], Xue Lei[5], Lu Wang[6], Hao Nie[1], Yuan Qi[7], Bin Tian [4], Alessandro Gardini [4], Andrew V. Kossenkov[8], Aaron Goldman [9], Shelley L. Berger [6], Ken-ichi Noma [2,3], Peter D. Adams [5] & Rugang Zhang [1] ✉

METTL3 is the catalytic subunit of the methyltransferase complex, which mediates m6A modification to regulate gene expression. In addition, METTL3 regulates transcription in an enzymatic activity-independent manner by driving changes in high-order chromatin structure. However, how these functions of the methyltransferase complex are coordinated remains unknown. Here we show that the methyltransferase complex coordinates its enzymatic activity-dependent and independent functions to regulate cellular senescence, a state of stable cell growth arrest. Specifically, METTL3-mediated chromatin loops induce Hexokinase 2 expression through the three-dimensional chromatin organization during senescence. Elevated Hexokinase 2 expression subsequently promotes liquid-liquid phase separation, manifesting as stress granule phase separation, by driving metabolic reprogramming. This correlates with an impairment of translation of cell-cycle related mRNAs harboring poly-methylated m6A sites. In summary, our results report a coordination of m6A-dependent and -independent function of the methyltransferase complex in regulating senescence through phase separation driven by metabolic reprogramming.

Cellular senescence is a stress response that is characterized by a stable cell growth arrest[1]. Senescence can be induced by triggers such as activation of certain oncogenes like RAS[1]. The stable growth arrest of senescent cells is accompanied by downregulation of cell cycle-promoting genes[2]. However senescent cells are metabolically active as exemplified by secretion of chemokines and cytokines, known as the senescence-associated secretion phenotype (SASP)[3,4]. These coordinated senescent alterations are essential for maintaining cellular homeostasis and facilitating physiological functions[5].

The catalytic methyltransferase-like 3 (METTL3) and the associated METTL14 are the core subunits of the methyltransferase complex (MTC) that mediates the RNA $N^6$-methyladenosine (m6A) modification[6]. m6A modification regulates gene expression in several different ways, including facilitating a liquid–liquid phase separation (LLPS)[7–9]. In addition, MTC regulates gene transcription in a m6A-independent manner through three-dimensional (3D) chromatin organization[10]. However, how these functions of MTC are coordinated at the molecular level remain unknown.

3D chromatin organization such as chromatin loops plays an important role in gene transcription[11]. For example, studies have demonstrated that physical convergence of enhancer and promoter activates gene transcription[12–14]. Interestingly, a genome-wide redistribution of the MTC promotes the expression of SASP genes[10]. Specifically, METTL3 is predominantly redistributed to pre-existing NFκB sites within the promoters of SASP genes. Although METTL3's localization to the enhancers is also increased, METTL14 is preferentially redistributed to these sites. Nevertheless, direct evidence of whether MTC affects 3D chromatin organization and its functional impact during senescence remains poorly understood.

Here we show that the MTC complex coordinates the senescence phenotypes by integrating its enzymatic activity-dependent and -independent functions. Specifically, METTL3-mediated chromatin loops induce Hexokinase 2 (HK2) expression independently of m6A modification, subsequently promoting phase separation in the form of stress granules during senescence. This phase separation inhibits the translation of cell-cycle related polymethylated m6A mRNAs, which contributes to senescence-associated stable growth arrest by preventing the death of senescent cells.

## Results

### Chromatin contacts promote MTC-regulated metabolic genes during senescence

To map genome-wide changes in chromatin interaction during senescence, we conducted H3K27Ac HiChIP analysis in control and senescent primary human lung fibroblasts IMR90 cells induced by an inducible oncogenic RAS (Fig. 1a and Supplementary Fig. 1a, b)[15]. The analysis revealed an overall decrease in the number of intra-chromosomal interactions in senescent compared with control cells (Supplementary Fig. 1c). To determine the potential role of MTC in the observed changes, we knocked down METTL3 or METTL14 in senescent cells (Supplementary Fig. 1d). Notably, knockdown of either METTL3 or METTL14 increased the number of interactions determined by H3K27Ac HiChIP analysis (Supplementary Fig. 1d). This suggests that MTC contributes to the observed decrease in chromatin contacts at a global level during senescence. We also classified the chromatin contacts into short-distance (<5Kb), medium-distance (5kb-2Mb), and long-distance (>2 Mb) categories. We found a decline in medium-distance interactions in senescence, which was largely rescued by knockdown of either METTL3 or METTL14 (Supplementary Fig. 1e and Supplementary Table 1).

We next determined whether alterations in chromatin contacts correlated with changes in transcription during senescence. Toward this goal, we performed fast global run-on sequencing (fastGRO-seq) to profile global nascent RNA transcription in proliferating and senescent cells (Fig. 1a)[16]. Indeed, there is a positive correlation between changes in chromatin contacts and gene transcription during senescence (Fig. 1b). This is particularly evident for those upregulated genes with increased chromatin contacts (Fig. 1b). Accordingly, we focused on the genes that are upregulated during senescence but downregulated upon knockdown of METTL3 or METTL14 (Fig. 1c, d). The analysis identified a list of 38 genes (Supplementary Table 2). We next subjected these genes to KEGG pathway enrichment. Consistent with the known role of MTC in regulating SASP[10], the analysis showed the enrichment of crucial pathways associated with the SASP, including NFκB signaling, chemokine signaling, and cytokine-cytokine receptor interactions (Fig. 1e and Supplementary Fig. 1f), which validated the experimental strategy we used in the present study.

Interestingly, the analysis revealed an enrichment of metabolic pathway, exemplified by HK2 expression (Fig. 1f and Supplementary Fig. 1g). Indeed, HK2 is upregulated in senescent compared with control cells and the observed upregulation was rescued by knockdown of METTL3 or METTL14 (Supplementary Fig. 1d). This was not a consequence of overcoming senescence because markers of senescence

such as expression of p21 and markers of cell cycle exit such as downregulation of cyclin A were not affected by knockdown of METTL3 or METTL14 (Supplementary Fig. 1d). As a negative control, knockdown of METTL3 or METTL14 did not affect the expression of 40 highest expressed genes in senescent cells based on fastGRO-seq analysis (Supplementary Fig. 1h), highlighting the specific regulation of the SASP and metabolic genes by the MTC during senescence.

To identify those MTC-regulated genes that are mediated by chromatin contacts, we cross-referenced the 38 genes with H3K27Ac HiChIP dataset. The analysis revealed a list of ten genes with enhanced chromatin contacts in senescent cells whose upregulation was METTL3/14 dependent (Fig. 1g). The list includes five SASP genes and five metabolic genes including HK2. This is consistent with previous reports that senescence is associated with metabolic reprogramming and enhanced energetic needs[17,18]. Strikingly, we observed a significant positive correlation between the observed increase in these chromatin contacts and fold changes in gene expression (Fig. 1g). Together, these data support that an increase in chromatin contacts promotes expression of MTC-regulated SASP and metabolic genes such as HK2.

### MTC transcriptionally activates HK2 through chromatin looping during senescence

HK2 mediates the rate-limiting step of glucose metabolism[19]. Despite HK2's functional importance in senescence[17,18], its regulation in senescence remains poorly understood. To unravel the regulatory mechanism by which the MTC promotes HK2 expression, we first cross-referenced publicly available H3K27Ac ChIP-seq, METTL3, and METTL14 Cut&Run-seq[10,20]. The analysis revealed an increased intensity of bound peaks along the HK2 genomic locus for H3K27Ac, METTL3, and METTL14 at either promoter or putative distal enhancer sites in senescent compared with control cells (Fig. 2a), suggesting that the observed upregulation occurs at the transcription levels mediated by enhancer-promoter interaction. To validate the observed HK2 upregulation occurs at the transcription level, we performed the KAS-seq, a chemical-based genetic approach that captures transcriptional activation with high sensitivity[21]. When integrating KAS-seq with fastGRO-seq and pol II ChIP-seq, the peak signals were consistently increased at the HK2 locus in senescent compared with control cells, which was decreased by knockdown of METTL3 or METTL14 (Fig. 2b). This result further supports that MTC is required for transcriptionally activation of HK2.

We next determined whether chromatin loops play a role in the observed upregulation of HK2. Toward this goal, we analyzed significant chromatin loops within −200 kb upstream and +100 kb downstream of HK2 gene body using H3K27Ac HiChIP. Two strong loops were newly formed in senescent cells spanning the HK2 genomic locus (Fig. 2c). Notably, knockdown of METTL3 or METTL14 restored the loop distribution as observed in control cells (Fig. 2c). This suggests that MTC may play a direct role in the formation of observed chromatin loops in senescent cells. To test this possibility, we conducted METTL3 HiChIP-seq. Similar to H3K27Ac HiChIP, we observed a reduction in intra-chromosomal loops during senescence (Supplementary Fig. 2a). Likewise, we noted a decrease in the percentage of medium-distance loops and an increase in the percentage of long-distance loops (Supplementary Fig. 2b and Supplementary Table 3). A detailed examination of loop formation based on METTL3 HiChIP analysis demonstrated an identical loop spanning HK2 genomic locus as shown in H3K27Ac HiChIP that was newly formed in senescent cells (Fig. 2c, d). We have previously shown that METTL3 regulates SASP gene expression by localizing to the pre-existing NF-κB genomic sites through the interaction with p65, which is known as a transcription factor that directly binds to DNA[10]. To explore the mechanism of how METTL3 affects high-order chromatin structure, we cross-referenced previously published p65 Cut&Run -seq and METTL3 Cut&Run-seq along the HK2 genomic locus and revealed an increased p65 bound

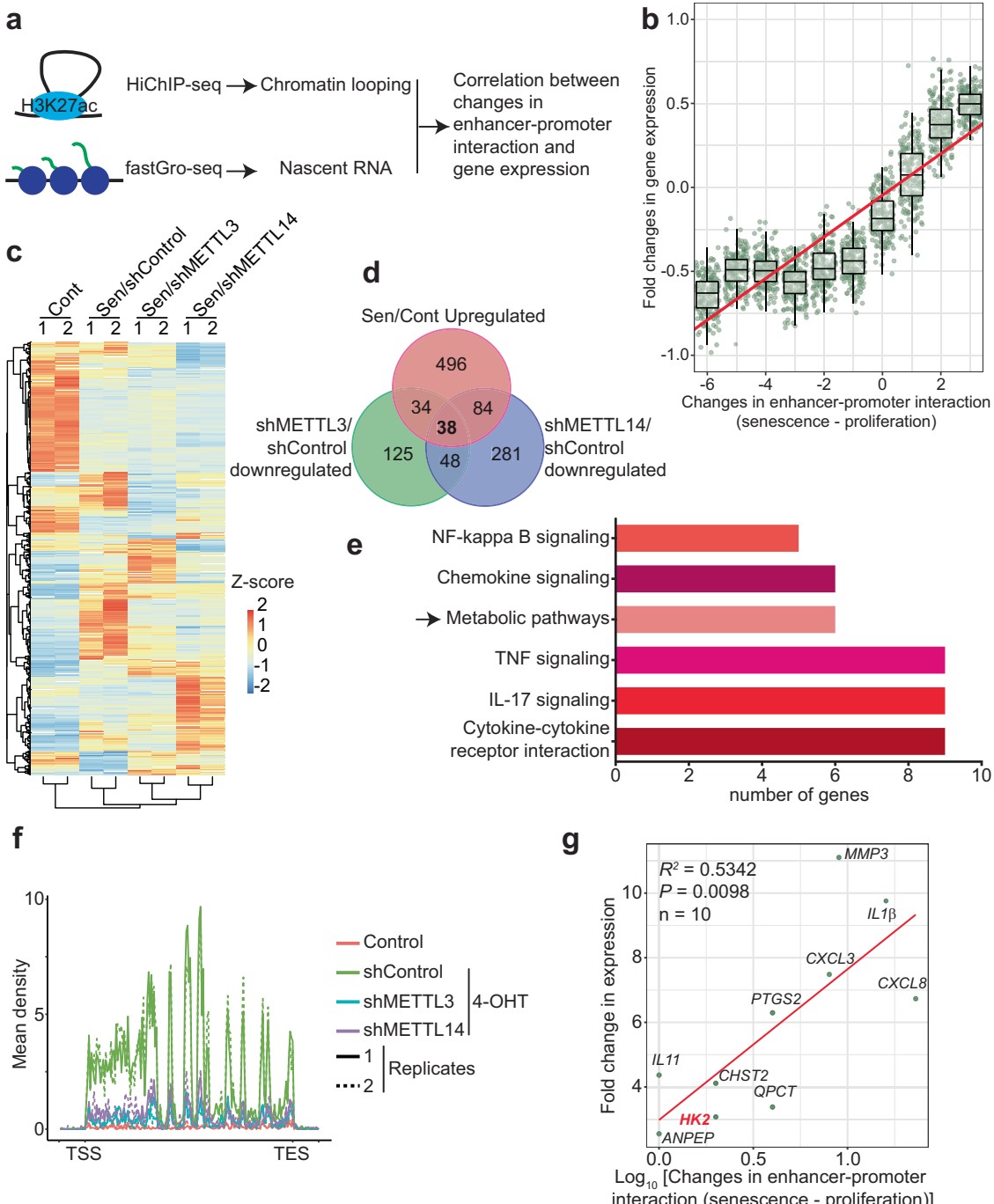

**Fig. 1 | Chromatin contacts promote MTC-regulated genes during senescence.**
**a** Schematic diagram showing genomic approaches employed: HiChIP-seq for monitor chromatin looping and fastGRO-seq for tracking nascent RNA transcription. **b** Pearson correlation between differences in chromatin contact numbers and alterations in nascent transcription in senescent cells relative to proliferating cells. $n = 2$ biologically independent fastGRO-seq experiments. The bottom and top edges of the box plot respectively represent the 25th and 75th percentiles, while the center refers to the median values of fold changes in gene expression. The whiskers extend to the minimum and maximum values within 1st and 99th percentiles range, respectively. **c** Heatmap showing profiles of nascent RNA transcription in proliferating and senescent cells with or without knockdown (KD) of METTL3 or METTL14. **d** Venn diagram showing overlapping of genes under indicated conditions. **e** KEGG pathway enrichment analysis showing enriched 40 genes depicted in (**d**) with pathways associated with SASP and metabolic process. **f** Mean density profiles of fastGRO reads for metabolic genes under indicated conditions. **g** Pearson correlation coefficient of 10 MTC-regulated genes exhibiting enhanced chromatin contacts was calculated between changes in chromatin contacts and differences in nascent transcription. $P$ value was calculated using two-sided Pearson r analysis.

peaks at putative distal enhancer in senescent compared to control cells, indicating that METTL3 may regulate high-order chromatin structure through its association with p65 (Supplementary Fig. 2c). In addition, to explore whether chromatin RNA methylation was involved in the loop formation of HK2, we analyzed chromosome-associated regulatory RNAs (carRNAs) in control and senescent cells with or without METTL3 and METTL14 knockdown using our previously published carRNA m6A-seq[10]. We showed that HK2 was not subject to the regulation of chromatin RNA methylation at putative distal enhancer anchors (Supplementary Fig. 2d).

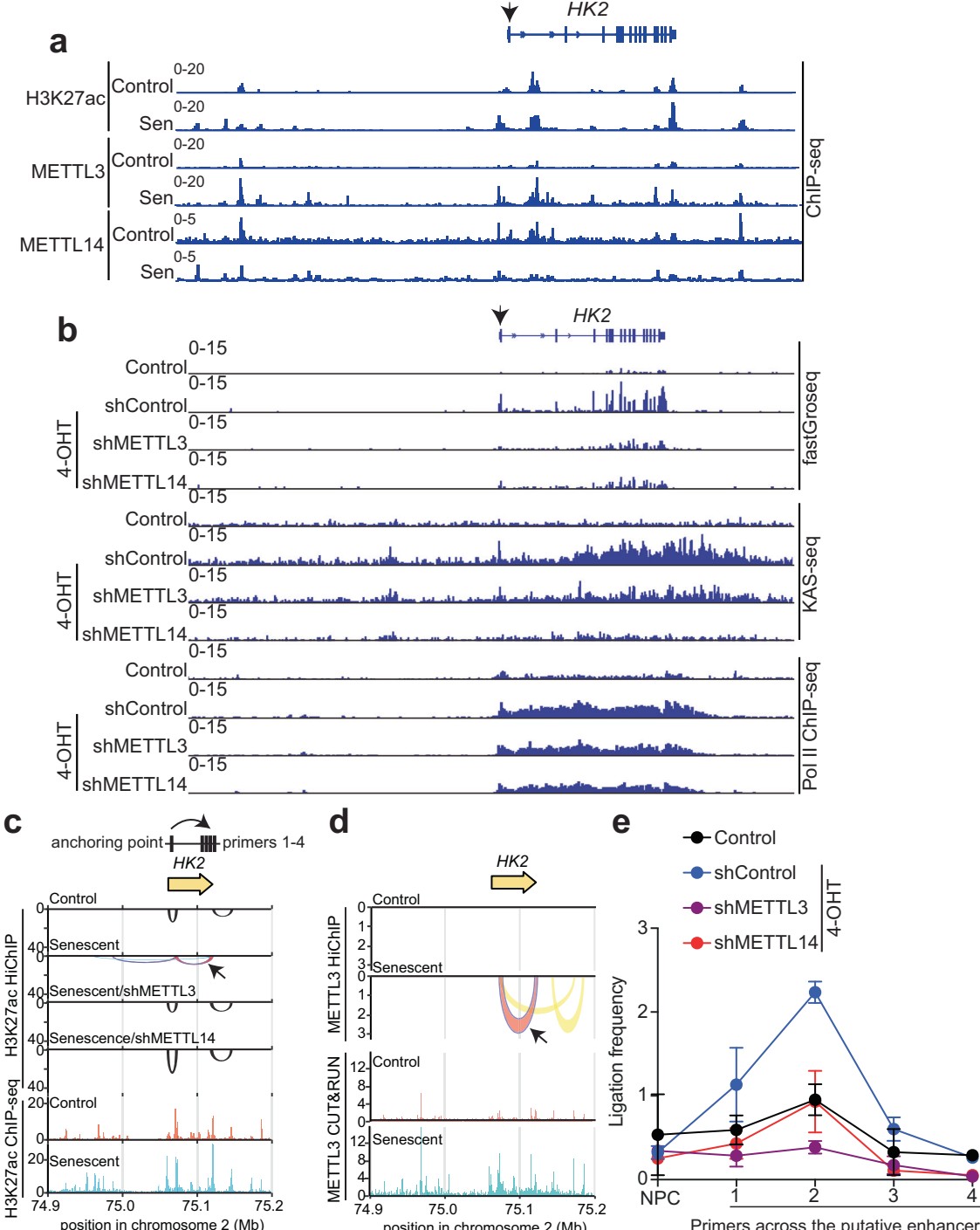

**Fig. 2 | HK2 upregulation by MTC depends on the chromatin looping during senescence. a** Tracks of publicly available H3K27Ac ChIP-seq (GSE74328), METTL3, and METTL14 Cut&Run-seq (GSE141992) showing peaks along *HK2* genomic region in RAS-induced senescent cells and proliferating control cells. Arrow indicates the transcription start site (TSS) of *HK2*. **b** Tracks of fastGRO-seq, KAS-seq, and publicly available Pol II ChIP-seq (GSE141992) showing peaks surrounding *HK2* genomic locus in proliferating and senescent cells with vector, METTL3, or METTL14 KD. Senescence was induced by 4-hydroxy-tamoxifen (4-OHT) in ER: RAS-expressing IMR90 cells. Arrow indicates the TSS of *HK2*. **c** Chromatin loops surrounding the *HK2* locus indicated by H3K27ac HiChIP in proliferating and senescent cells with either vector, METTL3, or METTL14 KD. Tracks of publicly available ChIP-seq of H3K27Ac (GSE74328) in proliferating and senescent cells displayed peaks aligned

the loop location as shown in H3K27Ac HiChIP. **d** Chromatin loops at the *HK2* locus revealed by METTL3 HiChIP in proliferating and senescent cells, aligned with tracks of publicly available METTL3 Cut&Run-seq (GSE141992). Arrow indicates the loop from METTL3 HiChIP coinciding with the loop shown in (**c**). **e** 3C-qPCR analysis of the ligation frequency on the indicated *HK2* gene loci in proliferating and senescent cells with either vector, METTL3 or METTL14 KD. 3C-qPCR primers were designed as illustrated in Fig. 2c with one forward primer anchoring at the promoter of *HK2* and the reverse primers targeting the downstream enhancer region of *HK2* according to the ChIP-seq peaks for H3K27Ac in control and senescent cells. NPC, non-peak control. Data represent the mean from *n* = 2 biologically independent experiments. Source data are provided as a Source Data file.

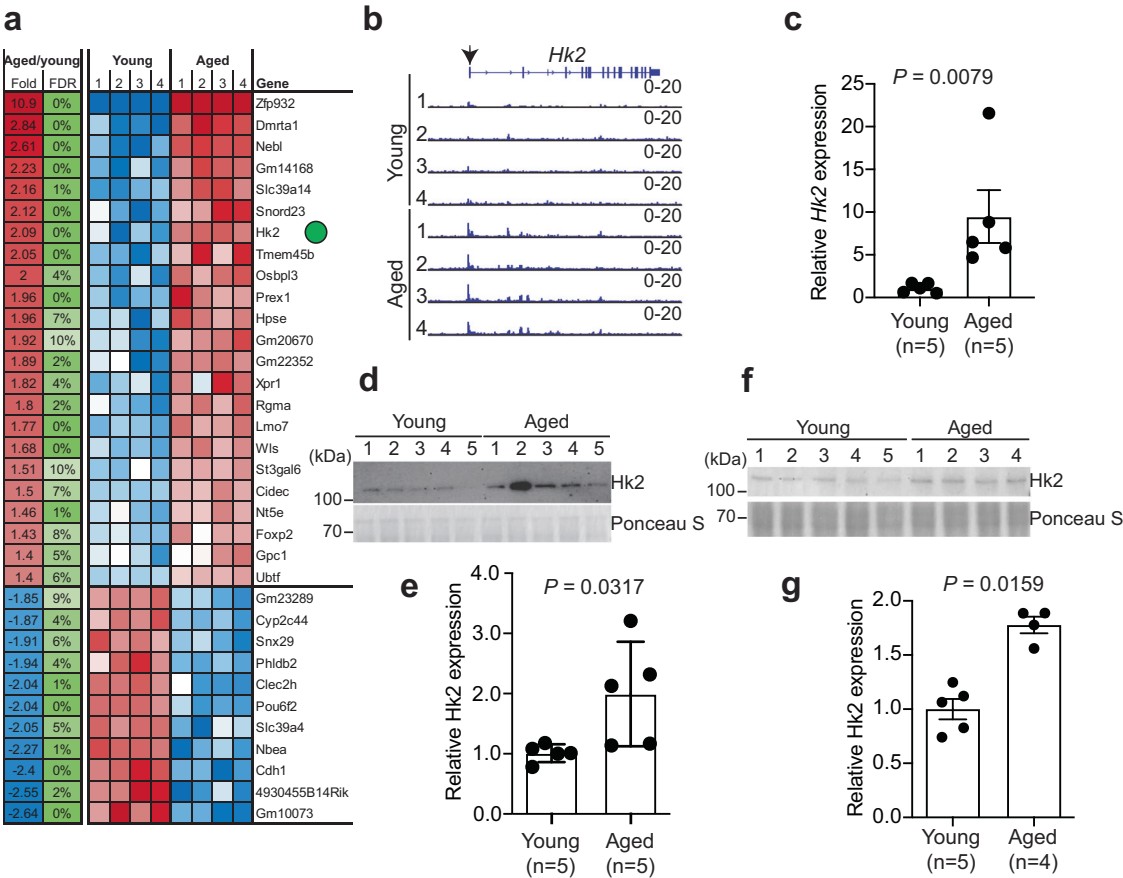

**Fig. 3 | Hk2 is upregulated during mouse liver aging. a** Heatmaps showing quantitative analysis of ATAC-seq data in young and aged whole liver tissues with a false discovery rate (FDR) cutoff of <0.1. Red represents higher expression, while blue represents lower expression. Data represent two technical replicates of two biological replicates. **b** Tracks displaying ATAC-seq peak signals in aged liver tissues compared to the young. Arrows indicate the transcription start site of *Hk2*. **c** RT-qPCR analysis of *Hk2* expression in young ($n = 5$) and aged ($n = 5$) hepatocytes. Data represent mean ± SD using a two-tailed Mann Whitney test. **d** Western blot analysis showing the expression level of Hk2 in young ($n = 5$) and aged ($n = 5$) whole liver tissues. Ponceau S staining was used as the internal control. **e** The band intensity of Hk2 compared to Ponceau S, as in (**d**), was quantified. Data represent mean ± SD using a two-tailed Mann Whitney test. **f** Western blot analysis showing the expression level of Hk2 in isolated hepatocytes from young ($n = 5$) and aged ($n = 4$) mice. Ponceau S staining was used as the internal control. **g** The band intensity of Hk2 compared to Ponceau S, as in (**f**), was quantified. Data represent mean ± SD values using a two-tailed Mann-Whitney test. Source data are provided as a Source Data file.

Functional topologically associated domains (TADs), which govern chromatin interactions within specific regions, have been shown to spatially regulate gene expression including those mediated by chromatin loops[22]. By leveraging previously published chromosome conformation capture (Hi-C) studies, we detected a gain of two sub-TAD domains surrounding *HK2* genomic locus with enhanced interaction frequencies in senescent compared with control cells (Supplementary Fig. 2e)[23]. The location of acquired sub-TADs aligns consistently with the loops formed in the H3K27Ac HiChIP, indicating enhanced interactions in restricted chromatin domains around *HK2*.

To directly examine chromatin looping for *HK2* gene locus guided by the common new loop formed in senescent cells based on H3K27Ac and METTL3 HiChIP, we performed chromosome conformation capture quantitative PCR (3C-qPCR) analysis in control and senescent cells[24]. We observed robust interactions between the anchored transcription starting sites (TSS) and the putative enhancer of *HK2* in senescent cells (Fig. 2e). Notably, knockdown of METTL3 or METTL14 reduced the association to the levels observed in control cells (Fig. 2e). In addition, restoration of enzymatic dead mutant METTL3 was sufficient to rescue HK2 expression, further supporting the notion that METTL3 regulates HK2 in a m⁶A independent manner (Supplementary Fig. 2f). Indeed, we found no significant change in m⁶A levels within the *HK2* genomic locus (Supplementary Fig. 2g). Consistently, treatment

of senescent cells with STM2457, a highly selective inhibitor of METTL3 methyltransferase activity[25], did not decrease *HK2* expression (Supplementary Fig. 2h, i). Moreover, knock down of YTHDC1, a nuclear reader of m⁶A, did not affect HK2 expression[26] (Supplementary Fig. 2j, k). Together, these findings suggest that HK2 is transcriptionally activated independently of m⁶A modification, which correlates with METTL3-mediated chromatin looping at the *HK2* locus in senescent cells.

### *Hk2* is upregulated during tissue aging

Senescent cells accumulated in aged tissues and contributed to tissue aging[27]. Thus, we sought to examine whether *Hk2* is regulated during tissue aging. Toward this goal, we employed tissue transposase-accessible chromatin with sequencing (ATAC-seq) to evaluate chromatin accessibility in liver tissues from young and aged mice (Fig. 3a). The analysis revealed that *Hk2* exhibited a more open chromatin status in the liver of aged compared with young mice (Fig. 3a). And a detailed examination of the ATAC-seq peaks demonstrated enhanced peak signals specifically at the promoter region of *Hk2* (Fig. 3b). Consistently, hepatocytes RNA-seq confirmed an elevated expression level of *Hk2* in aged compared to young mice, which was further validated by RT-qPCR (Fig. 3c and Supplementary Fig. 3a). Likewise, the protein level of Hk2 was upregulated in aged compared to young mice in both

whole liver tissues and isolated hepatocytes (Fig. 3d–g). In addition, the expression level of *Mettl3* and *Mettl14* remains no change in aged mice hepatocytes (Supplementary Fig. 3b). Finally, mining the Aging Atlas database confirmed *HK2* upregulation in multiple other aged tissues (Supplementary Fig. 3c)[28]. Together, we conclude that HK2 is upregulated during tissue aging.

## HK2 promotes purine metabolism during senescence

The two HK2 catalyzing domains convert glucose to Glucose-6-phosphate (G-6-P) that is a rate-limiting manner in glycolysis[29,30]. Point mutations at both N-terminal and C-terminal catalytic domains (D209A and D657A) of HK2 effectively diminish the enzymatic activity of HK2[29,31]. To explore the metabolic consequences of HK2 upregulation in senescence, we conducted $^{13}$C glucose isotope tracing in control and senescent cells with or without HK2 knockdown, and then rescued with ectopic expression of either wildtype or enzymatic activity inactive mutant HK2 (Fig. 4a). Notably, metabolites involved in the glycolysis pathway, such as G-6-P, and Fructose 1,6-bisphosphate, displayed high labeling rates across each group (Supplementary Fig. 4a). Interestingly, the central intermediate metabolite Fructose 1,6-bisphosphate in the glycolysis pathway did not exhibit reduced labeling (M + 6) by HK2 knockdown (Supplementary Fig. 4a). This could be due to the compensatory increase in the isoenzyme HK1 level in response to HK2 KD (Supplementary Fig. 4b).

To unbiasedly identify metabolite products potentially regulated by the enzymatic function of HK2 during senescence, we employed the following criteria: a) showing more extensive labeling in senescent compared to control cells; b) exhibiting decreased labeling in senescent cells by HK2 knockdown; c) rescuing the decrease in labeling by expressing of wildtype HK2 but not by the mutant HK2. This analysis revealed that HK2 plays a major role in regulating metabolites involved in purine metabolism such as inosine monophosphate (IMP), adenosine monophosphate (AMP), adenosine diphosphate (ADP), adenosine triphosphate (ATP), adenosine and inosine (Fig. 4b–c). In addition, our results showed that Ribose-5-phosphate (R-5-P) displayed lower $^{13}$C labeling in senescent compared to control cells (Fig. 4d). Consistently, the pentose phosphate pathway is known to provide nucleotide precursors for purine metabolism by branching from G-6-P and producing the intermediate product R-5-P[32]. Indeed, the level of R-5-P labeling was increased by HK2 knockdown, which was rescued by wildtype HK2, but not the mutant HK2 (Fig. 4d). In addition, the level of ATP and inosine that were reduced by METTL3 knockdown could be rescued by HK2 restoration, suggesting that METTL3's function in purine metabolism depends on HK2 (Supplementary Fig. 4c–e). Together, our data support a model whereby HK2 promotes purine metabolism and ATP production during senescence in an enzymatic activity-dependent manner (Fig. 4e).

## HK2 promotes phase separation during senescence

ATP is required for stress granules formation and dynamics[33]. These membrane-less granules can result from LLPS that is promoted by polymethylated m[6]A RNAs along with m[6]A readers such as YTHDF proteins[7–9,34]. MTC is essential for the formation of these endogenous phase-separated compartments, which correlates with m[6]A modifications[35]. Inspired by HK2-depenent increase in ATP levels observed during senescence, we proceeded to investigate whether the assembly of phase-separated stress granules occur during senescence. Accordingly, we stained for YTHDF1-3 and TIAR as phase separation and stress granules markers, respectively[8]. We found that 5–8% of senescent cells undergo phase separation in the form of stress granules as determined by colocalization of YTHDFs and TIAR (Fig. 5a–c). Comparable percentages of phase-separated senescent cells were observed for all three YTHDF family members (Fig. 5c and Supplementary Fig. 5a, b). Supporting the notion that the observed colocalized YTHDF and TIAR indeed represent phase-separation. 1,6-hexanediol, an organic chemical known to reverse phase separation[36], significantly decreased the number of senescent cells with the colocalization (Supplementary Fig. 5c, d). As a negative control, no phase separation was observed in RAS-transformed NIH3T3 cells (Supplementary Fig. 5e). This indicates that the observed phase separation resulted from senescence induction instead of RAS expression.

We next sought to determine whether the observed phase separation depends on METTL3-regulated HK2 upregulation in senescent cells. We knocked down HK2 or METTL3 in senescent cells. Indeed, knockdown of either HK2 or METTL3 led to a marked reduction in the phase separation observed in senescent cells (Fig. 5d, e). Similar observation was also made using a BRAF-induced senescent model (Supplementary Fig. 6a, b). Additionally, we exposed control proliferating cells to NaAsO$_2$, a known inducer of phase-separated stress granules in nearly 90% of cells[8]. Consistently, knockdown of HK2 or METTL3 was sufficient to reduce the phase-separated events induced by NaAsO$_2$ (Supplementary Fig. 6c, d). Thus, METTL3-regulated HK2 is required for formation of phase-separated stress granules.

To determine whether the observed phase separation depends on the enzymatic activity of the upregulated HK2 in senescent cells, we compared phase separation events in HK2 knockdown cells restored with either wildtype or an enzymatic activity inactive mutant HK2. Indeed, wildtype HK2 was significantly more efficient in restoring the reduction in stress granule formation induced by HK2 knockdown in senescent cells compared with the mutant HK2 (Fig. 5f, g). Consistently, treatment of senescent cells with 2-Deoxy-D-glucose, a HK2 inhibitor[37], effectively reduced stress granules formation in senescent cells (Supplementary Fig. 6e, f). In addition, we showed that there was no colocalization of HK2 with stress granules during senescence (Supplementary Fig. 6g). Human hexokinase isoenzymes include four genes named HK1 to HK4[19]. While HK3 and HK4 expression levels were barely detectable in senescent cells, we asked whether Hexokinase 1 (HK1) inhibition could diminish phase separation in senescent cells. Consistent with the notion that HK2-regulated phase separation is its enzymatic activity dependent, HK1 knockdown also reduced the phase-separated stress granules in senescent cells, albeit to a much lesser degree compared to HK2 knockdown (Supplementary Fig. 6h, i). This could be due to the fact that HK1 exhibits activity with only its C-terminal domain, whereas HK2 contains two catalytically active domains[29]. Collectively, these findings indicated that HK2 promotes phase separation during senescence in an enzymatic activity-dependent manner.

Since phase separation observed in the form of stress granules depends on MTC activity[35], we determined whether inhibition of METTL3 enzymatic activity affects stress granule formation. Indeed, treatment of senescent cells with STM2457, a selective inhibitor of METTL3 enzymatic activity, significantly reduced stress granules formation in senescent cells (Fig. 5h, i). This result indicates that phase separation observed in senescent cells in the form of stress granules depends on m[6]A modification.

## HK2 modulates translation of cell-cycle-related polymethylated m[6]A mRNAs

Stress granules impact mRNA fate in various ways, including a reduction in their stability and translation[7]. mRNAs that contain multiple, but not single, m[6]A sites have been shown to promote the formation of phase-separated stress granules[7,9]. To determine the functional consequences of phase separation during senescence, we focused on mRNAs with polymethylated m[6]A signals in senescent cells (Fig. 6a). The analysis of m[6]A-seq revealed 5223 unique m[6]A sites among genomic regions including 5' untranslated region (5' UTR), coding sequence, and 3' UTR in senescent cells (Supplementary Fig. 7a). Notably, 523 mRNAs containing more than three m[6]A sites were identified, which we defined as polymethylated m[6]A mRNAs in senescent cells. Since stress granules reduce mRNA stability[7], we focused on those polymethylated m[6]A transcripts that are downregulated in

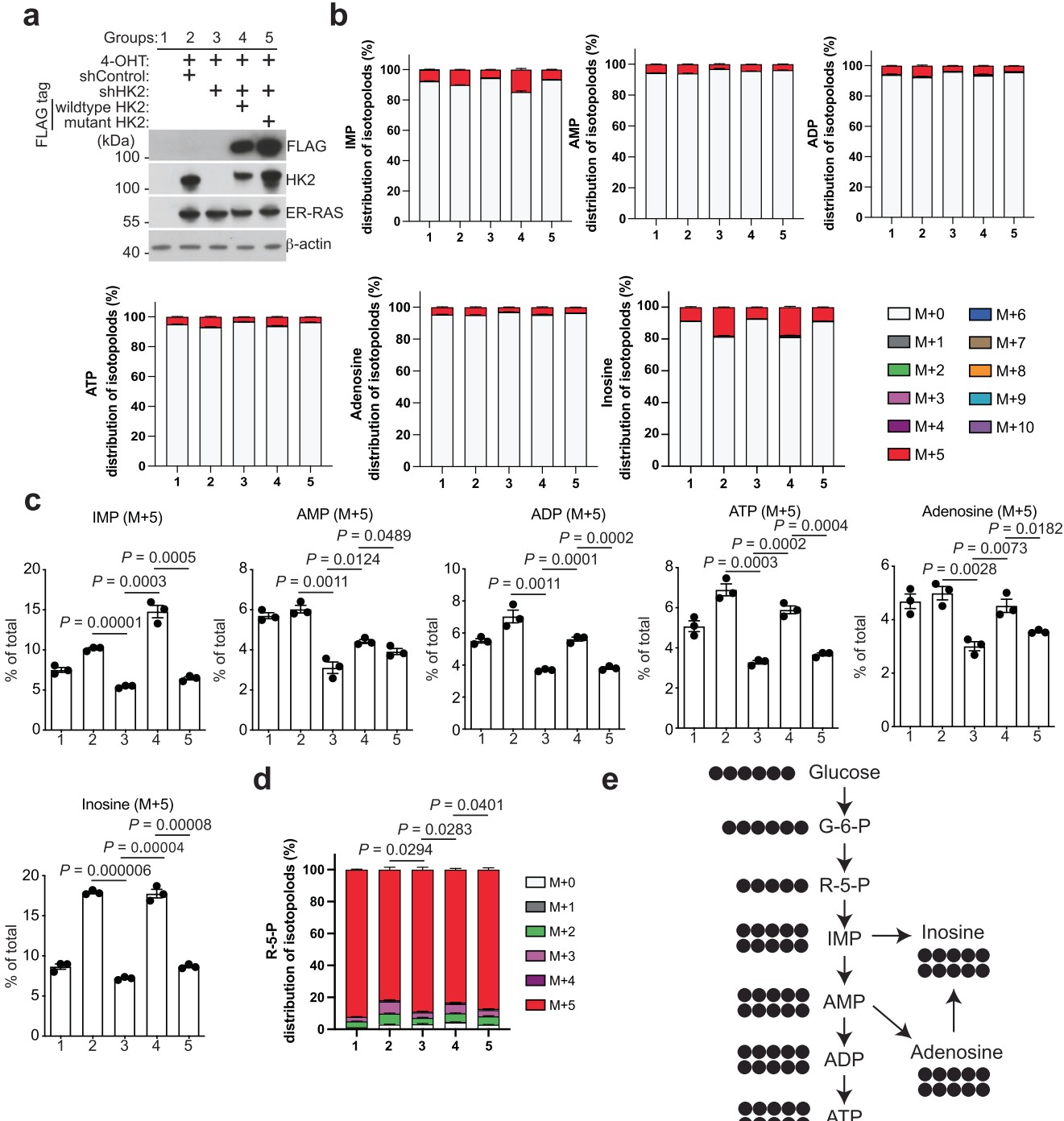

**Fig. 4 | HK2 promotes purine metabolism during senescence. a** Five groups were analyzed by ¹³C glucose isotope tracing. 1, control proliferating cells. 2, RAS-induced senescent cells via 4-OHT with control shRNA (shControl). 3, senescent cells with shRNA targeting HK2 (shHK2). 4, senescent cells with shRNA targeting HK2 (shHK2) and rescued with wildtype Flag-HK2. 5, senescent cells with shRNA targeting HK2 (shHK2) and rescued with mutant Flag-HK2. Note that the HK2-overexpressing plasmid is resistant to shHK2 due to silent mutation. IMR90 cells with or without 4-OHT induction of RAS expressing shControl or shHK2 with or without rescue by ectopic expression of wildtype or catalytic-mutant HK2 (D209AD657A) were analyzed for expression of the indicated proteins by western blot on Day 9. **b** Distribution of ¹³C isotopologues of Inosine monophosphate (IMP), Adenosine monophosphate (AMP), Adenosine diphosphate (ADP), Adenosine diphosphate (ATP), Adenosine, and Inosine at indicated groups. Data represent mean ± SEM of $n = 3$ biologically independent experiments. **c** Boxplots showing the labeled fractions of IMP (M + 5), AMP (M + 5), ADP (M + 5), ATP (M + 5), Adenosine (M + 5), and Inosine (M + 5) under indicated conditions. Data represent mean ± SEM of $n = 3$ biologically independent experiments. *P* values were calculated using a two-tailed Student's *t* test. **d** Distribution of ¹³C isotopologues of Ribose-5-phosphate (R-5-P) at indicated groups. Data represent mean ± SEM of $n = 3$ biologically independent experiments. *P* values were calculated using a two-tailed Student's *t* test. **e** Diagram showing labeled metabolites through ¹³C glucose isotope tracing. Black circles represent ¹³C atoms. G-6-P Glucose-6-phosphate, R-5-P Ribose-5-phosphate, IMP inosine monophosphate, AMP Adenosine monophosphate, ADP Adenosine diphosphate, ATP Adenosine triphosphate. Source data are provided as a Source Data file.

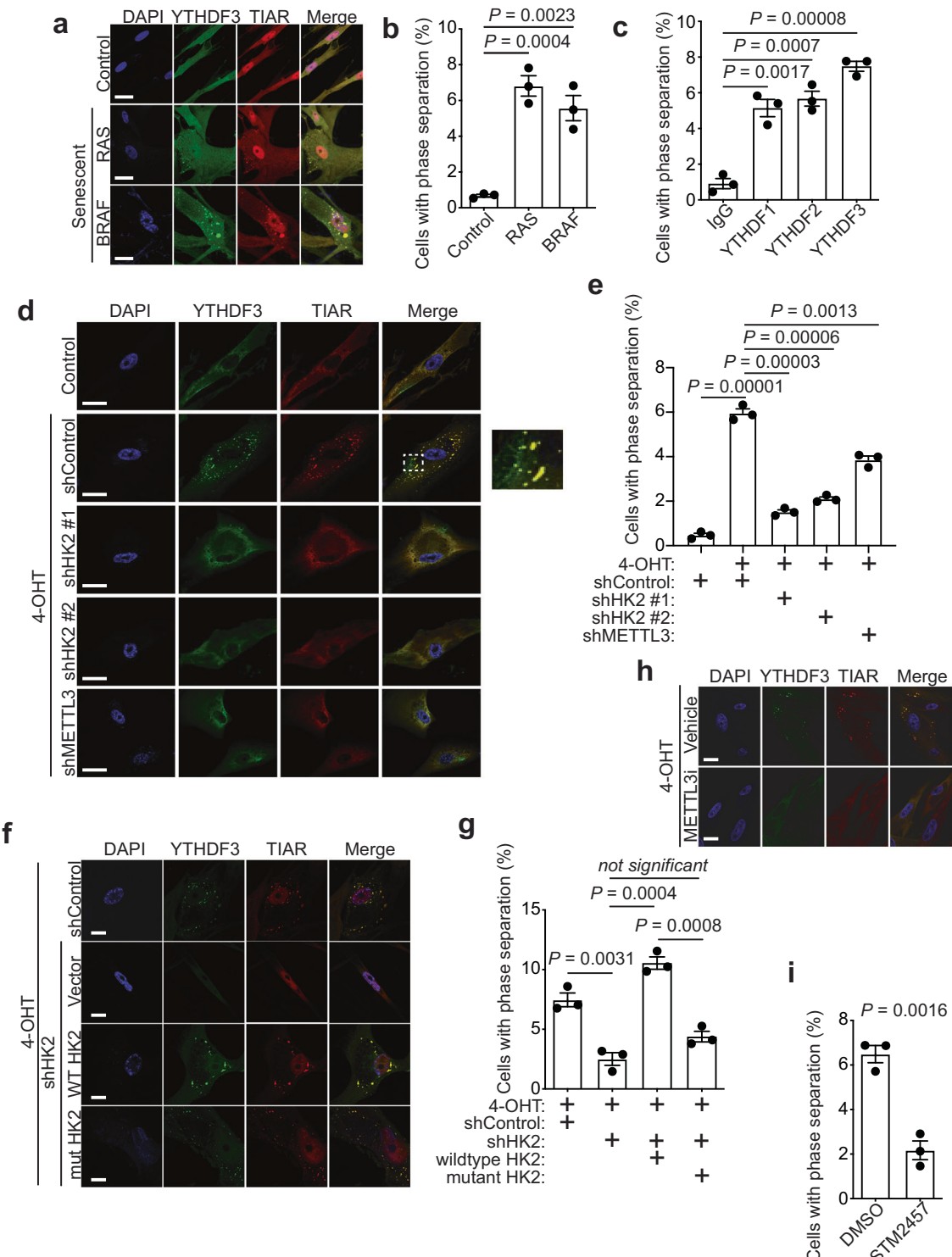

senescent compared with control cells. This analysis revealed a list of 35 polymethylated m⁶A mRNAs. Pathway analysis showed that genes encoded by these transcripts are significantly enriched for pathways such as cell cycle, cell division, and mitosis (Fig. 6b, c and Supplementary Fig. 7b).

To explore the potential physiological relevance of the newly identified polymethylated m⁶A mRNAs, we analyzed the published m⁶A-seq from liver or heart tissues of young and aged primates[38]. Similar polymethylated m⁶A signals were detected for cell-cycle related genes such as *SMC2*, *SMC3*, and *CCAR1* (Fig. 6d and Supplementary Fig. 7c, d). In addition, using publicly available single-cell RNA

sequencing (scRNA-seq) dataset from mouse skeletal muscle tissue[39], we found that the transcriptional abundance of cell-cycle related genes such as *Cdca8*, *Cenpe*, and *Smc2* exhibit lower levels in CD45⁻ senescent cells with high *HK2* expression compared with non-senescent cells (Fig. 6e and Supplementary Fig. 7e). As positive controls, SASP genes including *Cxcl1*, *Mmp2*, showed a similar pattern as *HK2* expression (Fig. 6e and Supplementary Fig. 7e). As a negative control, we found that *HK1* transcriptional level did not correlate with the senescent status (Supplementary Fig. 7e). Together, these results suggest that HK2 upregulation is associated with downregulation of cell cycle-related polymethylated m⁶A mRNAs.

**Fig. 5 | HK2 promotes phase separation during senescence in an enzymatic activity -dependent manner. a** Immunofluorescence (IF) of control and oncogene (RAS or BRAF)-induced senescent cells with antibodies targeting YTHDF3 and TIAR. 4′,6-diamidino-2-phenylindole (DAPI) counterstaining was used to visualize the nuclei. **b** Quantification of the percentage of cells with phase separation events at indicated conditions, as in **a**, was determined ($n > 200$ cells over three independent experiments). $P$ values were calculated using a two-tailed Student's $t$ test. Error bars represent mean with SD. **c** Quantification of the percentage of cells with phase separation events by antibodies targeting control IgG, YTHDF1, YTHDF2, and YTHDF3 in RAS-induced senescent cells was determined ($n > 200$ cells over three independent experiments). $P$ values were calculated using a two-tailed Student's $t$-test. Error bars represent mean with SD. **d** IF of control and RAS-induced senescent cells with antibodies targeting YTHDF3 and TIAR at indicated conditions. DAPI counterstaining was used to visualize the nuclei. Rectangle represented the area that was shown on the right with 5X magnification, showing cases of phase separation. **e** Quantification of the percentage of cells at indicated conditions, as in (**d**), was determined ($n > 200$ cells over three independent experiments). $P$ values were calculated using a two-tailed Student's $t$ test. Error bars represent mean with SD. **f** IF of RAS-induced senescent cells with antibodies targeting YTHDF3 and TIAR at indicated conditions. DAPI counterstaining was used to visualize the nuclei. **g** Quantification of the percentage of cells with phase separation events at indicated conditions, as in (**f**), was determined ($n > 200$ cells over three independent experiments). $P$ values were calculated using a two-tailed Student's $t$ test. Error bars represent mean with SD. **h** IF of RAS-induced senescent cells with antibodies targeting YTHDF3 and TIAR when treated with DMSO or METTL3 inhibitor (2.5 µM STM2457) for 48 h. **i** Quantification of the percentage of cells with phase separation events at indicated conditions, as in (**h**), was determined ($n > 200$ cells over three independent experiments). $P$ values were calculated using a two-tailed Student's $t$ test. Error bars represent mean with SD. Scale bars, 20 µm. Source data are provided as a Source Data file.

We next determined the role of HK2 in regulating translation of cell-cycle related polymethylated m⁶A mRNAs by performing polysome profiling in senescent cells with or without HK2 knockdown. Indeed, HK2 knockdown significantly increased the translational efficiency of polymethylated m⁶A mRNAs such as *CDCA8*, *CENPE*, and *SMC2* (Fig. 6f and Supplementary Fig. 7f). Since HK2 upregulation promotes phase-separated stress granules that impair the translation of polymethylated m⁶A mRNAs[7], we further knocked down YTHDFs to disrupt the phase separation in senescent cells (Supplementary Fig. 7g). Consistently, knockdown of YTHDFs increased translational efficiency for polymethylated m⁶A mRNAs such as *SMC2* and *CENPE* in senescent cells (Supplementary Fig. 7h). As a control, knockdown of YTHDF3 did not increase the translational efficiency for transcripts such as *CDCA8*, *CENPE*, and *SMC2* before HK2 upregulation (Supplementary Fig. 7i, j). These findings support that HK2 suppresses the translation of cell-cycle related polymethylated m⁶A mRNAs through phase-separated stress granules.

We next determined the consequence of increased translation efficiency of cell-cycle related mRNAs induced by HK2 knockdown. We found that HK2 knockdown increased the percentage of senescent cells in the G₂/M phase (Supplementary Fig. 8a) and induced a concurrent decrease in cell viability (Supplementary Fig. 8b, c). Supporting the notion that the observed effects are due to HK2's role in regulating phase-separated stress granules, similar observation was also made by disrupting stress granules through knockdown of YTHDFs (Supplementary Fig. 8d, e). To directly determine whether the observed decrease in cell viability was due to the death of senescent cells, we stained these cells with SPiDER-β-Gal to identify senescent cells together with a live/dead cell marker. Indeed, both knockdown of HK2 and YTHDFs increased the percentage of dead senescent cells compared to controls (Fig. 6g, h and Supplementary Fig. 9). Together, these data support a model whereby HK2 contributes to senescent-associated stable cell growth arrest by preventing the death of senescent cells through phase-separated stress granules.

## Discussion

We observed a global decrease in chromatin contacts as determined by both H3K27Ac and METTL3 HiChIP analysis. Knockdown of METTL3 restored the overall decrease in chromatin contacts. However, an increase in enhancer-promoter interaction positively correlates with changes in transcription in senescent cells. This suggests a redistribution instead of a simple decrease of chromatin contacts during senescence. Indeed, in the present studies, we showed that an increase in chromatin contacts correlates with upregulation of SASP and metabolic genes. These findings are consistent with previously reported genome-wide redistribution of MTC during senescence[10]. HK2 is functionally important in modulating glycolysis[19]. Here, we unveiled a direct link between METTL3-mediated chromatin interactions and the upregulation of HK2 during senescence. HK2

upregulation was also observed in aged hepatocytes and other aged tissues. Mechanistically, HK2 metabolically increases purine metabolism during senescence, which is consistent with previously published evidence that senescent cells have higher HK2 activity, exhibit enhanced glycolysis and produce more ATP[17,18]. This finding also aligns with previous observations of elevated purine metabolism in aged drosophila[40]. Notably, senescent cells are not dividing. This suggests that the observed increase in purine metabolism and the associated production of ATP may play a role in maintaining the senescent status. Consistent with this notion, inhibition of HK2 triggers the death of senescent cells. This finding indicates that HK2-regulated purine metabolism could be a target for developing senolytics to eliminate senescent cells. In addition, ATP plays a key role in regulating the formation and dynamics of stress granules[33,41]. Indeed, we showed that the enzymatic activity of HK2 is required for formation of stress granules in senescent cells. Similar to HK2 inhibition, disruption of stress granules genetically by knocking down YTHDFs induces the death of senescent cells. Notably, the regulation of polymethylated m⁶A mRNAs by phase-separated stress granules only occurs in senescent but not proliferating cells. This is because despite the presence of polymethylated m⁶A mRNA, there is no stress granules in proliferating cells without senescence-associated metabolic reprogramming. Collectively, these findings support the key role of HK2-dependent metabolic reprogramming in preventing the death of senescent cells through inducing phase-separated stress granules.

We observed 5−8% phase-separated stress granules in senescent cells. There are several possibilities that may account for this relatively low percentage. For example, this could be due to the dynamic nature of stress granules assembly and the inherent heterogeneity among senescent cells[1,42]. In addition, it is possible that a higher frequency of these phase-separated events could be observed using more sensitive techniques[8,33]. However, the mechanism we discovered in the present study may be broadly applicable. For example, we showed that METTL3 and HK2 are necessary for formation of stress granules that are chemically induced in ~90% of cells. Regardless, these findings advanced our conceptual understanding by which senescent cells leverage phase-separated stress granules to counteract stress via metabolic reprogramming.

We showed that MTC transcriptionally upregulates *HK2*, which correlated with chromatin looping during senescence. Likewise, MTC transcriptionally promotes the SASP genes in an enzymatic activity-independent manner[10]. In addition, HK2 prevents the death of senescent cells by inducing the formation of phase-separated stress granules in an m⁶A-dependent manner. Thus, MTC coordinates two key phenotypes of senescence, namely the senescence-associated growth arrest and the SASP, by integrating its enzymatic activity -dependent and -independent functions. These results are consistent with the notion that metabolic needs and SASP are intimately linked[17] and the MTC may represent one layer of the integration between these two

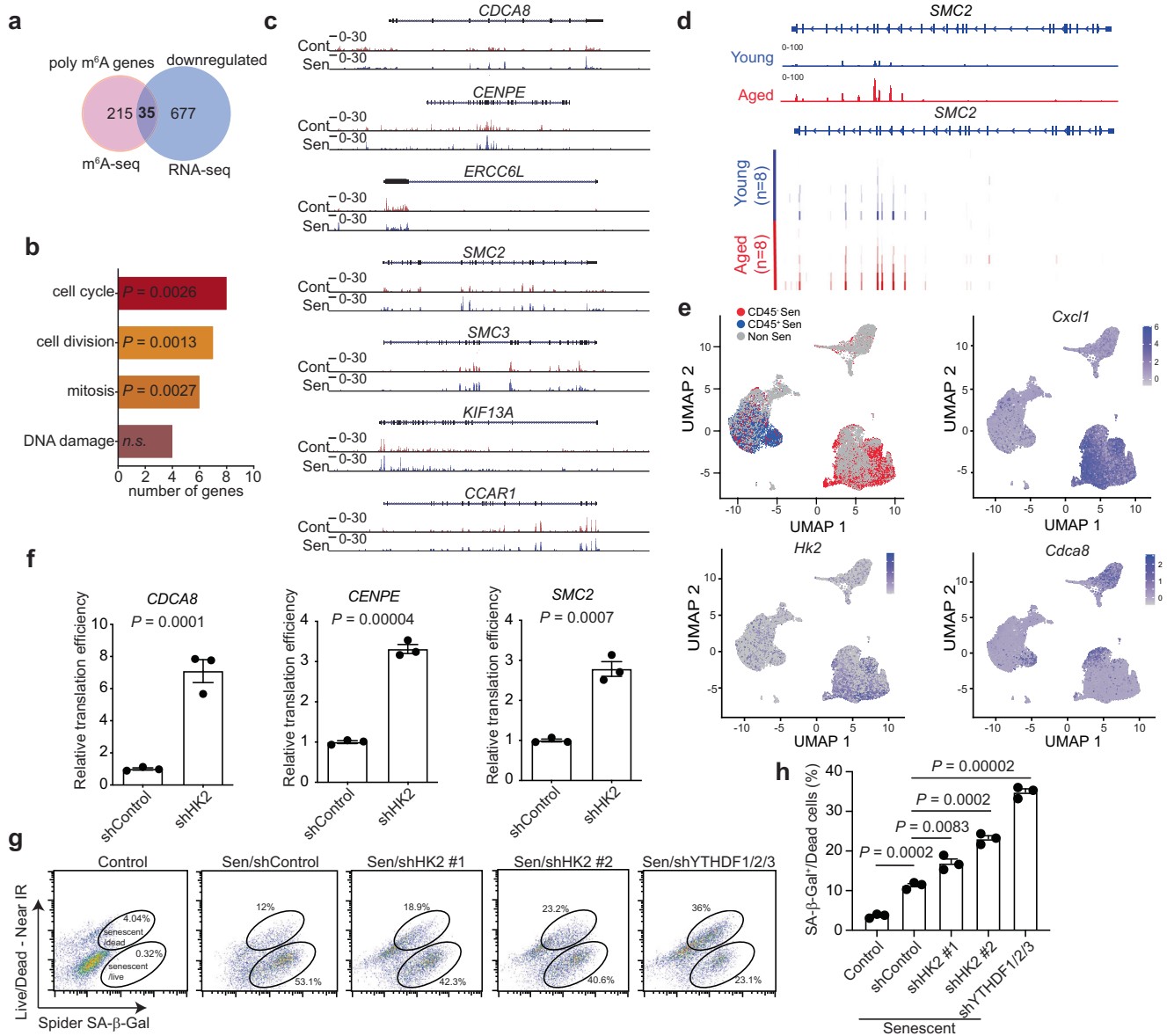

**Fig. 6 | HK2 maintains stable cell growth arrest of senescent cells by modulating cell-cycle related mRNAs with polymethylated m⁶A sites. a** Venn diagram illustrates the enrichment of RNAs with increased polymethylated m⁶A signals from publicly available m⁶A-seq when comparing senescent to control cells (GSE141993), overlapping with downregulated transcripts from publicly available RNA-seq (GSE141991). The highest m⁶A peak signal among all polymethylated m⁶A signals was used for comparison. **b** Pathway enrichment analysis showing enriched 35 genes depicted in (**a**) with pathways associated with cell cycle, cell division, and mitosis. **c**, Tracks of publicly available m⁶A-seq (GSE141993) from control (Cont) and senescent (Sen) cells showing peaks within the genomic locus of *CDCA8*, *CENPE*, *ERCC6L*, *SMC2*, *SMC3*, *KIF16A* and *CCAR1*, indicating polymethylated m⁶A signals. The m⁶A signal was normalized to the corresponding input. **d** Tracks (upper panel) of publicly available m⁶A-seq (CRA005942) in primates showing the median m⁶A peak signals within the *SMC2* genomic locus from either the young or aged group. Heatmap (lower panel) revealed the distribution of the m⁶A peak signals

from young (*n* = 8) and aged (*n* = 8) liver tissue in primates. The m6A signal was normalized to the corresponding input. **e** UMAP plot of publicly available scRNA-seq (GSE197017) from mouse skeletal muscle tissue showed clustering of cellular components of control (Non Sen) and senescent cells that were differentiated by CD45 marker (CD45+ Sen and CD45- Sen). UMAP projections were shown by Feature Plots depicting *Cxcl1*, *Hk2*, and Cdca8 expression in single cells. **f** Translational efficiency of mRNAs including *CDCA8*, *CENPE*, and *SMC2* was detected by polysome profiling. Translational efficiency is normalized to free mRNA level for each gene. Data represent mean ± SD of *n* = 3 biologically independent experiments. *P* values were calculated using a two-tailed Student's *t* test. **g** Senescent cells were co-stained by SPiDER SA-β Gal, and the LIVE/DEAD Near-IR dead cell stain kit then analyzed by flow cytometry under indicated conditions. **h** Quantification of dead senescent cells at indicated conditions, as in (**g**). Data represent mean ± SD of *n* = 3 biologically independent experiments. *P* values were calculated using a two-tailed Student's *t* test. Source data are provided as a Source Data file.

phenotypes of senescent cells. In summary, our findings show that senescent cells employ phase-separated stress granules via METTL3-mediated HK2 upregulation to govern the senescence-associated stable growth arrest by preventing the death of senescent cells. This mechanism attributes to the coordination of both m⁶A-independent and -dependent functions of METTL3 during senescence.

# Methods

## Ethics

All animal procedures were performed by protocols and guidelines approved by the Institutional Animal Care and Use Committee (IACUC) at the Sanford Burnham Prebys (SBP) Medical Discovery Institute.

## Cell culture and senescence induction

IMR90 primary human diploid lung embryonic fibroblasts (ATCC, CCL-186) were cultured in DMEM medium supplemented with 10% fetal bovine serum (FBS), l-glutamine, sodium pyruvate, non-essential amino acids, sodium bicarbonate and 1% penicillin−streptomycin under low oxygen tension (2%). For the experiments, only IMR90 cells that had undergone 25−36 population doublings were used unless otherwise stated. Viral packing cells 293T (ATCC, CRL-3216) or Phoenix (a kind gift from Dr. Gary Nolan, Stanford University) was cultured in DMEM medium supplemented with 10% FBS and 1% penicillin−streptomycin. Mouse NIH 3T3 fibroblasts (a kind gift from Dr. Bin Tian of The Wistar Institute) were cultured in DMEM medium supplemented with 10% calf serum, sodium pyruvate, and 1% penicillin−streptomycin. Cell lines were authenticated at The Wistar Institute's Genomics Facility using short-tandem-repeat DNA profiling. Regular mycoplasma testing was performed using PCR detection as previously described.

Induction of senescence by oncogenic BRAF$^{V600E}$ was carried out using the previously described protocol[43]. For ER: RAS-induced senescence, IMR90 cells were infected with retrovirus containing a 4-hydroxytamoxifen (4-OHT)-inducible ER: RAS construct (pLNC-ER: Ras)[44]. After 2 weeks of selection with G418 (400 µg ml$^{-1}$, Gibco), cells were maintained with a lower dose of G418 (200 µg ml$^{-1}$) and treated with 4-OHT at a final concentration of 100 nM. Cells were then harvested on Day 6 to examine the expression of RAS and other senescence markers.

## Reagents, plasmids and antibodies

METTL3 inhibitor STM2457 was purchased from MCE (HY-134836), Cycloheximide was purchased from Thermofisher (J66901.03), and NAC was ordered from Sigma (A9165).

The pBABE-puro- BRAF$^{V600E}$ and pBABE-puro-Empty plasmids were obtained from Addgene (#15269 and #1764). The template for HK2-3XFlag was a gift from Jason Cantor (Addgene plasmid # 163451). A PCR reaction was performed using primers that contained complementary sequences to the BamHI/EcoRI restriction enzyme sites, and the resulting PCR products were purified using a Zymo DNA purification kit (D4013). The purified PCR products were then ligated into a BamHI/EcoRI-digested PCDH-CMV vector (Addgene #72265) using the ClonExpress Ultra One Step Cloning Kit (Collagen C115-01) to obtain PCDH-CMV-HK2-3XFlag according to the manufacturer's protocols. To generate the HK2-expressing plasmid that is resistant to HK2 KD, Mut Express II Fast Mutagenesis Kit V2 (C214) was used according to the manufacturer's instructions. Site-specific mutation of HK2 enzymatic pocket (D209AD657A) was generated using the same kit. All plasmid construction and mutation sites were validated by Sanger sequencing. The following antibodies were purchased from the indicated suppliers. For HiChIP: anti-H3K27Ac (Abcam #4729); anti-METTL3 (Abcam #195352). For immunofluorescence, anti-YTHDF1 (Proteintech, #17479-1-AP; 1:150); anti-YTHDF2 (Abcam, #245129; 1:150); anti-YTHDF3 (Proteintech, #25537-1-AP; 1:150); anti-TIAR (BD biosciences, #610352; 1:150); Alexa Fluor 488 conjugated Goat anti-Rabbit IgG (H + L) Cross-Adsorbed Secondary Antibody (Thermo, A-11008; 1:200);, Alexa Fluor 568 conjugated Goat anti-Mouse IgG (H + L) Cross-Adsorbed Secondary Antibody (Thermo, A11004; 1:200). For western blots: anti-RAS (Becton Dickinson, # 610001; 1:1000); anti-p21 (Abcam, # 7960; 1:1000); anti-β-actin (CST #4967; 1:5000); anti-METTL3 (Abcam # 195352; 1:1000); anti-METTL14 (Sigma, #HPA038002; 1:1000); anti-HK2 (abcam, #209847; 1:1000), and anti-Flag (Sigma, #F1804; 1:2000).

## Retrovirus and lentivirus packaging and infection

Retrovirus was packaged using Phoenix cells as previously described[10]. Lentivirus particles were generated for genetic knockdown, overexpression, and rescue experiments. Briefly, HEK293T cells were co-transfected with lentivector-containing plasmids, including pLKO.shscramble (Addgene, #1864), pLKO.shMETTL3 (TRCN0000034715), pLKO.shMETTL14 (TRCN0000015936), pLKO.HK2-1 (TRCN0000037670), pLKO.HK2-2 (TRCN0000037671), pLKO.YTHDF1 (TRCN0000062768), pLKO.YTHDF2 (TRCN 0000167813), pLKO.YTHDF3 (TRCN0000370250), pLKO.HK1-1 (TRCN0000037655), pLKO.HK1-2 (TRCN0000037657), shYTHDC1-1 (TRCN0000243987), shYTHDC1-2 (TRCN0000243989) along with the packaging plasmids pMD2.G and psPAX2 (Addgene #12260 and Addgene #12259) using the cationic polymer polyethylenimine (PEI) at a ratio of 1:3. After 48 and 72 h of transfection, the lentiviral particles were harvested from the culture supernatant by filtering through a 0.45 µm filter (Corning, 431220). The harvested lentivirus was aliquoted and stored at −80 °C until further use. Senescent IMR90 cells were infected with shRNA-encoding lentivirus with 8 µg/ml polybrene (Sigma, #28728-55-4). 2 µg/ml Puromycin selection was used to select knockdown cell lines for 48−72 h after transduction with lentiviral particles.

## HiChIP and loop calling

HiChIP was performed mainly according to an established procedure[15]. Briefly, 15 million of IMR90 cells were fixed in freshly made 1% formaldehyde at room temperature for 10 min followed by quenching with 125 mM glycine. Samples were lysed in ice-cold Hi-C lysis (10 mM Tris, pH 8, 10 mM NaCl, 0.2% IGEPAL CA-630 with protease inhibitors) buffer and digested with MboI (NEB, R1047) for 2 h at 37 °C. After 1 h of biotin dATP (Thermo, 19524016) incorporation, samples were ligated with T4 DNA ligase (NEB, M0202) at room temperature and ChIPped with H3K27Ac antibody (abcam, ab4729) or METTL3 antibody (abcam, ab195352) after sonication. IP samples were reversed-crossed, pulled down with streptavidin C1 beads (Invitrogen, 65001) and went through tagmentation by transposaseTn5 (Illumina, 20034198). Samples were amplified with certain cycle numbers according to qPCR with 5-cycle pre-amplified library. Libraries were purified through two-sided size selection with Ampure XP beads (Beckman Coulter, A63880) and were sequenced with 75-base pair-end reads on the Illumina NextSeq 500 platform. Raw fastq files were demultiplexed and HiC-Pro 3.1.0 was used to align the reads to the hg19 human genome as previously described. Reads were assigned to MboI restriction site and duplicated reads were removed. The following parameters were set: BOWTIE2_GLOBAL_OPTIONS = −very-sensitive, −end-to-end, −reorder, BOWTIE2_LOCAL_OPTIONS = −very-sensitive, −end-to-end, −reorder, GENOME_FRAGMENT = hg19_MboI.bed, LIGATION_SITE = GATCGATC. All other parameters were set as default. hichipper 0.7.7 was used for loop calling as previously described[45]. Library quality was determined, and chromatin loops were called using a restriction fragment-aware approach based on pre-determined narrow peaks. The narrow peak file of H3K27Ac ChIP-seq and METTL3 Cut&Run-seq were obtained from Gene Expression Omnibus (GEO; accession number GSE74328 and GSE141992). Following parameters were used: −peakpad 700 −make-ucsc. All other parameters were set as default. Loops that passed FDR < 0.01 with at least 4 reads and FDR < 0.05 were considered significant for H3K27Ac HiChIP and METTL3 HiChIP, respectively.

For loop mapping to promoters and enhancers, promoters were defined as 1 kb upstream and downstream from TSS. All significant HiChIP interactions overlapped with promoters for either side were defined as Enhancer-Promoter interaction. For correlation analysis of gene expression with looping, the total number of significant HiChIP interactions in growing (EP#(growing)) and senescent IMR90 cells (EP#(senescent)) were counted for each gene. Log fold-change of the fastGRO-seq scores were compared with the difference of significant interactions (EP#(growing) - EP#(senescent)). To avoid the effect of outliers, 70 genes were randomly selected from the same EP#(growing) - EP#(senescent) group 200 times and the average log fold-change was plotted.

## Hi-C analyses

The Hi-C data for both growing and senescent IMR90 cells were downloaded from GSE118494 and processed as previously described[23]. Briefly, the paired reads were individually mapped to the hg19 human genome using Bowtie2 (version 2.2.9) with the iterative alignment approach. The aligned reads were assigned to MboI fragments. Redundant paired reads derived from PCR biases, aligned to repetitive sequences, or with low mapping quality (MapQ <30) were filter out. Additionally, paired reads originated from self-ligation and undigested products were also eliminated. The human genome was divided into non-overlapped 10-kb bins. Raw contact matrices were constructed by counting paired reads assigned to each pair of bins, and Hi-C biases in the contact matrices were corrected using the Iterative Correction and Eigenvector decomposition (ICE) method[46]. The ICE normalization was repeated 30 times. The ICE-normalized contact scores were used for heat maps, where the maximum intensity corresponded to the top 5% score.

## fastGRO-seq

fastGRO-seq was performed mainly based on the published procedure[16]. Briefly, 20 million IMR90 cells at each condition were washed with ice-cold PBS and incubated with cold swelling buffer (10 mM Tris, pH 7.5, 2 mM $MgCl_2$, and 3 mM $CaCl_2$ with 2 U/ml Superase-in) for 5 min on ice. Cells were washed with GSB buffer (swelling buffer + 10% glycerol with 2 U/ml Superase-in) and lysed in lysis buffer (10 mM Tris pH 7.5, 2 mM $MgCl_2$, 3 mM $CaCl_2$, 10% glycerol, 1%l Igepal (NP-40), 2 U/ml Superase-in). Isolated nuclei were washed twice with cold lysis buffer and were frozen in freezing buffer (50 mMTris-HCL pH 8.0, 40% glycerol, 5 mM $MgCl_2$, 0.1 mM EDTA) at −80 °C. Drosophila spike-in nuclei were added to thawed nuclei before use. Freshly prepared and pre-warmed 2X Nuclear run-on buffer (10 mM Tris-HCl pH 8, 5 mM $MgCl_2$, 300 mM KCl, 1 mM DTT, 500 mM ATP, 500 mM GTP, 500 mM 4-thio-UTP, 2 mM CTP, 200 U/ml Superase-in, 1% Sarkosyl (N-Laurylsarcosine sodium salt solution)) was added and incubated for 7 min at 30 °C for the nuclear run-on. RNA was extracted and purified with TRIzol LS reagent (Invitrogen) and ethanol precipitation. Nuclear run-on RNA was then fragmented with a UCD-200 Bioruptor for 1–5 cycles of 30 s ON/30 s OFF at high settings. Fragmented RNA was incubated at 65 °C for 10 min then on ice for 5 min to ensure full exposing of 4-thio-UTP. Biotinylated solution (20 mM Tris pH 7.5, 2 mM EDTA pH 8.0, 40% dimethylformamide, 200 mg/ml EZ-link HPDP Biotin (Thermo Scientific)) was added to the fragmented RNA for 2 h at room temperature with shaking at 100 g. Followed by ethanol precipitation, the biotinylated RNA was pulled down by streptavidin M280 beads (Invitrogen). 4-thio-UTP labeled RNA was eluted in 100 mM DTT and purified through RNA Clean and Purification kit (Zymo Research). Genomic DNA was eliminated through in-column DNase procedure. Purified RNA was used to constructed RNA sequencing libraries using the NEBNext Ultra II Directional RNA Library Prep kit (New England Biolabs) and was sequenced on the Illumina NextSeq 500 system. Reads were aligned to the hg19 human genome and were processed and analyzed as previously described[16]. Bam files were normalized based on the drosophila spike-in RNA among each group and bigwig files were generated with deeptools 3.3.1. For average density analysis, deeptools was used to extract read densities, then mean density profiles were generated with R 4.2.0. using ggplot2[47]. For differential gene expression analysis, HOMER 4.10 was used to calculated RPKM (Reads Per Kilobase of transcript per Million fragments mapped) and DESeq2 1.12.3 was used to estimate the significance of differential gene expression among groups.

## Low-input KAS-seq

Low-input KAS-seq was performed mainly based on the published protocol[21]. Briefly, 10,000 IMR90 cells were incubated in media with 5 mM $N_3$-kethoxal for 10 min at 37 °C. Cells were pelleted and labeled genomic DNA (gDNA) was extracted by using Quick gDNA mini plus kit (Zymo, D4068). DNA was eluted with 25 mM $K_3BO_3$ (pH 7.0) and went through click reaction with 20 mM DBCO-PEG4-biotin (sigma 760749) in PBS solution. RNase A (Thermo, 12091039) was added and biotinylated gDNA was fragmented by Tn5 transposase (Illumina) at 37 °C for 30 min. Fragmented DNA was purified and eluted by DNA Clean & Concentrator-5 kit (Zymo, D4013) and was pulled down by streptavidin C1 beads (Invitrogen, 65001). DNA-conjugated beads and corresponding inputs were used for library construction by using index primers and purified by two-sided size selection with Ampure XP beads (Beckman Coulter, A63880). Generated library was sequenced in a 75-bp single-end mode on the Illumina NextSeq 500 platform and raw reads were demultiplexed. Reads were trimmed using trim-galore and low-quality reads or duplicates were removed. Aligned reads were extended to 150 bp to fit the average length of DNA fragments from the KAS-seq libraries. Bedtools were used to convert Bam files to bed files and bedGraph files. Bigwig files were generated by using ucsc-bedgraphtobigwig tools. MACS2 was used to call KAS-seq peaks using the following parameter: –broad -g hs –broad-cutoff 0.01 -q 0.01. All other parameters were set as default.

## ATAC-seq analyses

All procedures were performed by protocols and guidelines approved by the IACUC at the SBP Medical Discovery Institute. The mice were housed in a controlled environment with a 12-h light/dark cycle, while the ambient temperature was maintained at 22–23 °C with 40–60% humidity. Male young C57BL/6N (5-month-old, $n = 5$) and aged C57BL/6J (21-month-old, $n = 5$) mice were obtained from Charles River facility and fed by Teklad Gloabal 18% Protein Rodent diet (Inotiv/Envigo, ref 2018). Liver ATAC-seq were performed as previously described[48]. Briefly, whole liver was excised and maintained on ice cold DMEM and nuclei were isolated while tissue was fresh. Tagmentation was conducted for 30 min at 37 °C followed by DNA isolation and library preparation. For data analyses, reads were mapped to the mm10 mouse genome with Ensemble gene information. HOMER was used for peak calling, bigwig generation and peak counts quantification. 146,684 unique sites with significant peak in at least one sample were identified. DEseq2 was used with raw counts to estimate the significance between groups. Sites with at least one group showing peaks in all four replicates were considered.

## Hepatocytes RNA-sequencing analyses

All procedures were performed by protocols and guidelines approved by the IACUC at the SBP Medical Discovery Institute. Hepatocytes RNA from male young C57BL/6N (5-month old, $n = 5$) and aged C57BL/6J (21-month-old, $n = 5$) mice were extracted as previously described[49,50]. For data analyses, raw read quality was checked using FastQC v0.10.0. and reads were trimmed with a threshold quality score >20 using Trim-galore v0.3.0. Trimmed reads were then aligned to mm9 using bowtie2. Duplicated reads were marked using Picard Tools v1.98. Peaks were called using macs2 with parameter –nomodel.

## Western blot

Total protein extracts were prepared from indicated samples using RIPA buffer (50 mmol/L Tris-HCl (pH 8.0), 150 mmol/L NaCl, 1% Triton X-100, 0.5% sodium deoxycholate, 0.1% SDS, and 1 mmol/L phenylmethylsulfonyl fluoride) on ice for 20 min. Protein concentration was determined using a BCA protein assay kit (Pierce #23225). Equal amounts of protein samples (10–30 μg) were separated by SDS-PAGE using 8–15% polyacrylamide gels and transferred onto nitrocellulose membranes (Millipore). The membranes were blocked with 4% BSA (Sigma-Aldrich) in TBST buffer (Tris-buffered saline with 0.1% Tween-20) for 1 h at room temperature, and then incubated with specific primary antibodies against the target proteins of interest overnight at 4 °C. The membranes were washed three times with TBST buffer for 8 min each, followed by incubation with horseradish peroxidase-conjugated

secondary antibodies (CST, anti-rabbit #7074 and anti-mouse #7076) for 1 h at room temperature. Protein bands were detected using the Dura Extended Duration Substrate (Thermofisher, # 34075).

## Quantitative real-time PCR (qPCR)

Total RNA was extracted from cells using the Zymo Quick-RNA™ MiniPrep Kit (R1054) according to the manufacturer's instructions. RNA concentration and purity were determined using a NanoDrop One spectrophotometer (Thermo Fisher Scientific). Complementary DNA (cDNA) was synthesized from 1 µg of total RNA using the SuperScript™ VILO™ cDNA Synthesis Kit (Thermo Fisher Scientific, #11754050). qPCR was performed using iTaq Universal SYBR Green Supermix (Bio-Rad, #1725121) and run on the QuantStudio 7 Flex Real-Time PCR System. The cycling conditions included an initial denaturation at 95 °C for 3 min, followed by 40 cycles of denaturation at 95 °C for 10 s, annealing at 60 °C for 30 s, and extension at 72 °C for 30 s. The specificity of the PCR products was confirmed by melting curve analysis. Gene expression levels were normalized to the geometric mean of reference gene (GAPDH) using the comparative Ct (ΔΔCt) method, and fold changes were calculated using the $2^{-\Delta\Delta Ct}$ method. List of primers used in the present study is included in Supplementary Data 1.

## Metabolic tracing and measurements

Stable isotope tracer analysis using [$^{13}C_6$]-glucose was conducted with three biologically independent experiments at groups including control proliferating cells, RAS-induced senescent cells via 4-OHT with control shRNA (shControl), senescent cells with shRNA targeting HK2 (shHK2), senescent cells with shRNA targeting HK2 (shHK2) and rescued with wildtype Flag-HK2, and senescent cells with shRNA targeting HK2 (shHK2) and rescued with mutant Flag-HK2. The experiments were performed by seeding cells at a density of $3 \times 10^5$ cells per 6 cm dish and incubating them with 25 mM [$^{13}C_6$]-glucose tracer in glucose-free medium for 30 min. Following incubation, the cells were washed with chilled PBS and incubated with 500 µl of extraction solution (80:20 v/v methanol/water) at 4 °C for 5 min. Next, cells were scraped with a polypropylene cell scraper and the extraction solution from each sample was collected, vortexed, and incubated on dry ice for at least 30 min. Then, each sample was centrifuged at maximum speed at 4 °C for 10 min, and the resulting supernatant was used for analysis. Metabolite measurements were normalized based on the protein concentration in the protein pellets (Pierce BCA protein assay kit; # 23250). A sample pool was generated by mixing equal volumes of all extracts from unlabeled samples. LC–MS metabolite flux analysis was performed on a Thermo Scientific Q Exactive HF-X mass spectrometer equipped with heated electrospray ionization (HESI) II probe and coupled to a Thermo Scientific Vanquish UHPLC system. A Millipore SeQuant ZIC-pHILIC 2.1-mm i.d × 150 mm column with a gradient elution method was employed to separate the metabolites. The MS analysis of all samples was performed in full-scan, polarity-switch mode. The unlabeled sample pool was also analyzed by data-dependent MS/MS in positive and negative modes. The mass spectrometer was operated with a spray voltage set at 3.5 kV in positive ion mode and 3.2 kV in negative ion mode. The heated capillary and HESI probe were set at 325 °C and 350 °C, respectively, with the S-lens RF level set at 40. The gas settings for sheath, auxiliary, and sweep were 40, 10, and 2 units, respectively. Full MS scans were acquired for the top 10 highest abundance ions at a resolution of 15,000 with an AGC target of 5E4, a maximum IT of 50 ms, an isolation width of 1.0 $m/z$, and a stepped normalized collision energy of 20, 40, and 60. Annotation and quantitation of metabolites and carbon isotopologues with natural isotope abundance correction were performed using Compound Discoverer 3.3 software (Thermo Scientific).

## Immunofluorescence

For immunofluorescence staining, cells on coverslips were washed by phosphate-buffered saline (PBS) twice and fixed with 3.7% paraformaldehyde for 12 min at room temperature. The samples were washed twice with PBS and permeabilized with 0.2% Triton X-100 in PBS for 15 min. Following the PBS wash, the samples were blocked with 3% bovine serum albumin (BSA) in PBS for 20 min at room temperature and then incubated with the primary antibody diluted in dilution buffer (1% BSA in PBS) overnight at 4 °C. The next day, the samples were washed three times with PBST and probed with an Alexa Fluor-conjugated secondary antibody diluted in dilution buffer for 1 h at room temperature. The cells or tissues were washed and stained with 1 µg/ml 4,6-diamidino-2-phenylindole (DAPI) for 5 min at room temperature. After three washes with PBST, the coverslips or slides were mounted with an Prolong Gold anti-fade mountant (Thermofisher, #P36930) and imaged using a high-resolution confocal microscope (Leica TCS SP5 II).

## Colony formation assay

Cells were seeded in 24-well plates and cultured for 10 days before staining with 0.05% crystal violet for visualization. The cells were then washed with PBS and air-dried before being imaged using a high-resolution scanner. Images were processed using ImageJ software to quantify the integrated density of the stained cells.

## Flow cytometry

The cell-permeable SPiDER β-gal (Dojindo) was used for detection of senescent β-gal activity as instructed by the manufacturers' manual. Briefly, $5 \times 10^5$ cells of each sample were washed with Hanks' HEPES buffer twice and incubate at 37 °C for 15 min with 1 µmol/l SPiDER β-gal working solution. Following the incubation, LIVE/DEAD™ Fixable Near-IR Dead Cell Stain Kit (ThermoFisher L34975) was used to stain live or dead cells at RT for 15 min with protection from light. Cell suspensions were analyzed by flow cytometry using channels of FITC and Near-IR respectively.

## 3C-qPCR

The 3C-qPCR assays were performed following a previously described protocol[10]. Briefly, five million cells were fixed with 1% formaldehyde in fresh medium for 8 min at room temperature and quenched with 0.2 M glycine for 10 min. The cells were lysed in cold Hi-C lysis buffer (10 mM Tris pH 8.0, 10 mM NaCl, 0.2% IGEPAL CA-630, cOmplete protease inhibitor cocktail (Roche, # 11697498001)) for 15 min, followed by a wash with 500 µl cold lysis buffer. The cell pellet was resuspended in 500 µl 0.5% SDS and incubated at 62 °C for 10 min with shaking. Then, 150 µl water and 25 µl of 10% Triton X-100 were added to quench the SDS, and the samples were incubated at 37 °C for 15 min with shaking. Next, 25 µl 10× NEBuffer2 and 100 U Mbo I were added to digest the chromatin overnight at 37 °C with shaking. Following inactivation of the Mbo I at 62 °C for 20 min, the ligation was performed at room temperature for 4 h with shaking by adding 750 µl ligation master mix (100 µl of 10×NEB T4 DNA ligase buffer, 80 µl of 10% Triton X-100, 10 µl BSA (10 mg ml−1), 5 µl T4 DNA ligase (400 U µl−1), and 555 µl water). After centrifugation at 10,000 r.p.m. for 5 min, reverse cross-linking was performed at 65 °C for at least 4 h. The DNA was then purified by phenol–chloroform extraction and ethanol precipitation. Finally, qPCR was performed using iTaq Universal SYBR Green Supermix (Bio-Rad, #1725121) and run on the QuantStudio 7 Flex Real-Time PCR System. The data were analyzed using the comparative Ct method. List of primers used in the present study is included in Supplementary Data 1.

## Polysome profiling

Polysome profiling was performed according to the protocol described previously[51]. Briefly, $3 \times 10^7$ cells were washed with ice-cold PBS containing 100 mg ml−1 cycloheximide, harvested, and suspended in hypotonic buffer (10 mM HEPES pH 7.9, 1.5 mM MgCl2, 10 mM KCl, 200 U ml−1 SUPERaseIn RNase inhibitor, 1× protease inhibitor cocktail, and 0.5 mM dithiothreitol). The cells were then incubated on ice for

30 min and centrifuged at 14,000 $g$ for 10 min to remove the nuclear pellet. The resulting supernatant was loaded onto a 10 ml 10–50% sucrose gradient in a polyallomer tube containing 20 mM HEPES–KOH pH 7.5, 15 mM $MgCl_2$, 80 mM KCl, 2 mM dithiothreitol, and 100 mg ml⁻¹ cycloheximide, and centrifuged in a TH-641 rotor (Thermo Fisher) at 19,000 $g$ at 4 °C for 2 h. The gradient was fractionated using a system consisting of a syringe pump (Harvard Apparatus model 11), a density gradient fractionator (Brandel), and an ISCO UA-6 UV/VIS detector. Polysome-free, monosome, and polysome fractions were separated based on their UV-light absorbance. The monosome and polysome fractions were incubated with 25 mM EDTA (pH 8.0), 10 mM Tris–HCl (pH 7.0), and 1% SDS at 65 °C for 5 min. The RNA was then extracted with phenol–chloroform and ethanol precipitation. cDNA was obtained and qPCR was performed following the abovementioned protocols.

## Quantification of N6-methyladenosine (m⁶A) in RNA

The m⁶A RNA methylation status was detected using total RNA (200 ng) isolated from samples with or without treatment of the METTL3 inhibitor STM2457 (2.5 µM) by the EpiQuik m⁶A RNA Methylation Quantification Kit (Colorimetric) according to the manufacturer's protocols. Absorbance signals were measured on a microplate reader at 450 nm, and m⁶A RNA methylation levels were determined through background extraction.

## Measurement of ATP and inosine detection

ATP levels were detected in the indicated conditions using the ATP Luminescence Assay Kit (Dojindo laboratories) following the manufacturer's protocols. Inosine levels were measured under the indicated conditions by Inosine Quantification Assay Kit (ab126286) based on the manufacturer's procedures. Fluorescent signals were read at Ex/Em = 535/587 nm on a microplate reader.

## Statistics and reproducibility

Results are representative of a minimum of three biologically independent experiments with data expressed as mean ± standard deviation of the mean (SD) unless stated otherwise. Statistical analyses were conducted using GraphPad Prism 7 (GraphPad) or R 4.2.0. The significance of differences between groups was determined using a two-tailed unpaired Student's $t$ test, and $P$ values < 0.05 were considered statistically significant unless otherwise stated.

## Reporting summary

Further information on research design is available in the Nature Portfolio Reporting Summary linked to this article.

# Data availability

HiChIP-seq, fastGRO–seq, KAS-seq, ATAC-seq, and RNA-seq data and processed files generated in this study have been deposited in the GEO under the accession number GSE243047 (HiChIP-seq, fastGRO–seq, and KAS-seq), GSE243630 (ATAC-seq), and GSE243906 (RNA-seq). In addition, publicly available dataset such as GSE74328 (H3K27Ac ChIP-seq), GSE141992 (METTL3 and METTL14 Cut&Run-seq), GSE141993 (m⁶A-seq), GSE141994 (RNA-seq), GSE197017 (mouse skeletal muscle tissue scRNA-seq), GSE118494 (Hi-C), and CRA005942 (primates m⁶A-seq) were obtained from GEO. Metabolomics raw data has been deposited to Metabolomics Workbench (study_id: ST003182 [https://doi.org/10.21228/M8NQ82])⁵² Spectral data for relevant metabolomics can be found in Supplementary Data 2. Source data are provided with this paper.

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

## Acknowledgements

This work was supported by US National Institutes of Health grants R01CA160331 and R01CA276569 to R.Z., and P01AG031862 to K.N., P.D.A., S.L.B and R.Z. R.Z. is a CPRIT Scholar in Cancer Research. Support of Core Facilities was provided by Cancer Centre Support Grant (CCSG) CA010815 to The Wistar Institute and P30CA016672 to The University of Texas MD Anderson Cancer Center. Support of metabolomic study was provided by NIH grant U2C-DK119886 and OT2-OD030544 grants.

## Author contributions

C.W., H.T., C.H., A.H., Q.Z., L.L., X.H., X.L., L.W., H.N., Y.Q., A.V.K., and A.G. (Aaron Goldman) performed the experiments and analyzed data. C.W., R.Z. designed the experiments. A.H., Q.Z., X.L., L. W., A. G. (Ales-sandro Gardini), and B.T. contributed key experimental materials. S.B., P.D.A., K.N., and R.Z. supervised studies. C.W and R.Z. wrote the manuscript. R.Z. conceived the study.

## Competing interests

The authors declare no competing interests.

## Additional information

[1]Department of Experimental Therapeutics, University of Texas M.D. Anderson Cancer Center, Houston, TX 77030, USA. [2]Institute of Molecular Biology, University of Oregon, Eugene, OR 97403, USA. [3]Institute for Genetic Medicine, Hokkaido University, Sapporo 060-0815, Japan. [4]Gene Expression and Regulation Program, The Wistar Institute, Philadelphia, PA 19104, USA. [5]Sanford Burnham Prebys Medical Discovery Institute, San Diego, CA, USA. [6]Penn

Epigenetics Institute, Perelman School of Medicine, University of Pennsylvania, Philadelphia, PA, USA. [7]Department of Bioinformatics & Computational Biology, University of Texas MD Anderson Cancer Center, Houston, TX 77054, USA. [8]Immunology, Microenvironment and Metastasis Program, The Wistar Institute, Philadelphia, PA, USA. [9]Molecular and Cellular Oncogenesis Program, The Wistar Institute, Philadelphia, PA, USA.
✉e-mail: rzhang11@mdanderson.org

