## [Peer Review File · Nature Communications]

METTL3-mediated chromatin contacts promote stress granule phase separation through metabolic reprogramming during senescenceREVIEWER COMMENTS

Reviewer #1 (Remarks to the Author):

Methyltransferase-like 3 (METTL3) is traditionally thought to act as the catalytic subunit of the RNA N6-methyladenosine (m6A) methyltransferase complex to regulate target mRNAs and thus gene expression. In this paper, the Zhang laboratory studied into METTL3's enzymatic independent activity. Specifically, authors reported that METTL3 mediates the change in high-order chromatin structure during senescence of primary human lung fibroblasts (IMR90 cells), which is correlated well with activation of the underlying genes as shown by fastGRO-seq and H3K27ac/METTL3 Hi-CHIP. Among those activated genes (38) showing METTL3 dependence in senescent cells are enriched with cytokines, chemokines and NF- κ B targets, consistent with senescence associated secretion phenotype (SASP). Upregulation of these genes is independent of METTL3's enzymatic activity and is correlated with formation of new enhancer:promoter contacts. They then homed in on HK2, a rate limiting enzyme in glycolysis. HK2 is upregulated following the tissue aging, whose enzymatic activity enhances purine metabolism and ATP production during senescence. Authors further showed that the HK2 activation by METTL3 is crucial for stress granule phase separation (staining of YTHDF and TIAR), which acts to limit the translation of a set of poly-m6A marked mRNAs encoding for cell proliferation regulators. The observations thus support a model that HK2 contributes to senescence-associated stable cell growth arrest by inducing formation of phase-separated stress granules. Overall, this work is quite a tour de force using multiple current technologies. The data shown are generally solid and results are interesting. The topics also cover chromatin loop formation, phase separation, m6A and senescence, which would appeal to the field. The paper fits well into the journal and I would support its publication after authors address the below minor issues.

1/title: shall "phase separation" be specified as "stress granule phase separation"? Because phase separation exists very much widely, covering too many aspects of cellular processes that are not studied in the paper. Likewise, it is better to directly specify "stress granule phase separation" in the line 39 of abstract.

2/ How exactly does METTL3 enhance high-order chromatin structure? Does it bind DNA or RNA or cofactors at loop anchors? Authors shall at least speculate in the Discussion.

3/ Typo and grammar issues need to be fixed. For example:

Line 34, "remain" shall be "remains"; line 91: authors refer to Extended Data fig 1d (instead of 1a)? Line 207, "remain" shall be "remains"; In fig 6a, there are 250 poly-m6A mRNAs but the text mentioned 523 (line 209); etc. Please check out the manuscript again.

Reviewer #2 (Remarks to the Author):

In this manuscript the authors show that RNA N6-methyladenosine (m6A) methyltransferase active component METTL3 regulates transcription in an enzymatic activity-dependent and independent manner. It can modulate high-order chromatin structure, as previously found by the same group. This elevates HK2 transcription, which promotes liquid-liquid phase separation (LLPS) and drives metabolic reprogramming. The methyltransferase also methylated mRNA encoding cell-cycle related transcripts. The elevated HK2 appears to suppress translation of these methylated transcripts. These results depict an interesting over regulatory picture of MTC on metabolism reprogramming to affect cell senescence.

Most works are well done. I have three questions regarding the methylation part.

The MTC is known to mediate chromatin RNA methylation which regulates chromatin state as well. It would be nice if the authors could separate these effects. Maybe some complement experiments with wt and inactive mutant would help. Or to show no chromatin RNA methylation involved in these loop formation.

The authors could knock down YTHDC1, a nuclear reader of m6A, and presumably HK2 expression would not be affected.

HK2 might form stress granule and hijack YTHDF1 or YTHDF3, which are known to promote translation of methylated mRNA. I would suggest KD YTHDF1 or 3 and monitor translation efficiency change before HK2 activation. Overexpression of these genes may rescue the translation effect.

Reviewer #3 (Remarks to the Author):

In this study, the authors uncovered the functions of METTL3-mediated chromatin contacts in phase separation and metabolic reprogramming during senescence. The authors previously reported that m6A-independent genome-wide METTL3 and METTL14 redistribution drives the senescence-associated secretory phenotype (Nature Cell Biology 2021). Here they further showed that the MTC complex coordinates its enzymatic activity-dependent and -independent functions to regulate cellular senescence. In addition, METTL3-mediated chromatin loops induce Hexokinase 2 (HK2) expression through the three-dimensional chromatin organization during senescence. The elevated HK2 expression subsequently promotes liquid-liquid phase separation (LLPS) by driving metabolic reprogramming. Overall, this study provides the new mechanistic insights about the METTL3 enzymatic activity-dependent and -independent functions in regulating senescence. My specific comments are listed below:

Major:

1. Fig. 2a, given that the authors reported that METTL14 knockdown affect HK2, it's essential to address whether METTL14 also binds to HK2 promoter or enhancer?
2. Though the authors identified a list of 38 genes that are regulated by MTC complex during senescence, it remains to be explained that why those genes are specifically regulated by MTC complex.
3. The authors claimed that METTL3 promotes the transcription of HK2 independent of its catalytic activity by using the STM2457. In the original article (Reference 25), STM2457 was used to treat cells for a long time. Here the authors only treated the cells with low concentration for 48 hours, it remains to be validated that STM2457 did actually inhibit METTL3's enzymatic activity in the authors' experimental conditions.
4. Similarly, to prove that METTL3 promotes the transcription of HK2 independent of its catalytic activity, more direct evidence is that the catalytic mutant of METTL3 can promote HK2 transcription.
5. The authors concluded that METTL3 promotes HK2, which regulates purine metabolism. It remains to be demonstrated that whether METTL3 regulates purine metabolism, and whether METTL3's function in purine metabolism depends on HK2.

Minor :

1. "HK2 is upregulated in senescent compared with control cells and the observed upregulation was rescued by knockdown of METTL3 or METTL14 (Extended Data Fig. 1b)." Extended Data Fig. 1b is not the data the authors mentioned.
2. NaAsO₂, the 2 should be lower case?

We are very grateful to the reviewers for providing us with very helpful, consistent, and constructive feedback on our work, and for giving us the opportunity to revise and improve our manuscript.

We have conducted several new experiments, performed new analyses, and made several modifications throughout the manuscript and its figures to address all the reviewers' concerns. We have included detailed, point-by-point responses to each of these concerns, describing the corresponding changes in our manuscript. Our responses are highlighted in blue text to facilitate the review process.

Reviewer #1: Pages 1-2

Reviewer #2: Pages 3-4

Reviewer #3: Pages 5-7

REVIEWER COMMENTS

Reviewer #1 (Remarks to the Author):

Methyltransferase-like 3 (METTL3) is traditionally thought to act as the catalytic subunit of the RNA N6-methyladenosine (m6A) methyltransferase complex to regulate target mRNAs and thus gene expression. In this paper, the Zhang laboratory studied into METTL3's enzymatic independent activity. Specifically, authors reported that METTL3 mediates the change in high-order chromatin structure during senescence of primary human lung fibroblasts (IMR90 cells), which is correlated well with activation of the underlying genes as shown by fastGRO-seq and H3K27ac/METTL3 Hi-CHIP. Among those activated genes (38) showing METTL3 dependence in senescent cells are enriched with cytokines, chemokines and NF-kB targets, consistent with senescence associated secretion phenotype (SASP). Upregulation of these genes is independent of METTL3's enzymatic activity and is correlated with formation of new enhancer:promoter contacts. They then homed in on HK2, a rate limiting enzyme in glycolysis. HK2 is upregulated following the tissue aging, whose enzymatic activity enhances purine metabolism and ATP production during senescence. Authors further showed that the HK2 activation by METTL3 is crucial for stress granule phase separation (staining of YTHDF and TIAR), which acts to limit the translation of a set of poly-m6A marked mRNAs encoding for cell proliferation regulators. The observations thus support a model that HK2 contributes to senescence-associated stable cell growth arrest by inducing formation of phase-separated stress granules. Overall, this work is quite a tour de force using multiple current technologies. The data shown are generally solid and results are interesting. The topics also cover chromatin loop formation, phase separation, m6A and senescence, which would appeal to the field. The paper fits well into the journal and I would support its publication after authors address the below minor issues.

1/title: shall "phase separation" be specified as "stress granule phase separation"? Because phase separation exists very much widely, covering too many aspects of cellular processes that are not studied in the paper. Likewise, it is better to directly specify "stress granule phase separation" in the line 39 of abstract.

Response: We agree with the reviewer and have accordingly revised the title and abstract.

2/ How exactly does METTL3 enhance high-order chromatin structure? Does it bind DNA or RNA or cofactors at loop anchors? Authors shall at least speculate in the Discussion.

Response: We thank the reviewer for this insightful question. We have previously shown that METTL3 regulates SASP gene expression by localizing to the pre-existing NF- κ B genomic sites through the interaction with p65, which is known as a transcription factor that directly binds to DNA [1]. Accordingly, we cross-referenced p65 Cut&Run -seq and METTL3 Cut&Run-seq along the *HK2* genomic locus. The analysis revealed an increased p65 bound peaks at putative distal enhancer loop anchors in senescent compared to control cells, indicating that METTL3 may regulate high-order chromatin structure through its association with p65 (New data in **Extended Data Fig. 2c**). We have now included this result and the associated discussion in the revised manuscript.

3/ Typo and grammar issues need to be fixed. For example:

Line 34, “remain” shall be “remains”; line 91: authors refer to Extended Data fig 1d (instead of 1a)? Line 207, “remain” shall be “remains”; In fig 6a, there are 250 poly-m⁶A mRNAs but the text mentioned 523 (line 209); etc. Please check out the manuscript again.

Response: We thank the reviewer’s spotting these errors. We corrected the typos and grammar issues. In addition, the text mentioned 523 refers to “523 mRNAs containing more than three m⁶A sites in senescent cells”, while 250 poly-m⁶A mRNAs in fig 6a refers to “the RNAs with increased polymethylated m⁶A signals when comparing senescent to control cells”. We have now clarified this in the revised manuscript.

Reviewer #2 (Remarks to the Author):

In this manuscript the authors show that RNA N6-methyladenosine (m6A) methyltransferase active component METTL3 regulates transcription in an enzymatic activity-dependent and independent manner. It can modulate high-order chromatin structure, as previously found by the same group. This elevates HK2 transcription, which promotes liquid-liquid phase separation (LLPS) and drives metabolic reprogramming. The methyltransferase also methylated mRNA encoding cell-cycle related transcripts. The elevated HK2 appears to suppress translation of these methylated transcripts. These results depict an interesting over regulatory picture of MTC on metabolism reprogramming to affect cell senescence.

Most works are well done. I have three questions regarding the methylation part.

The MTC is known to mediate chromatin RNA methylation which regulates chromatin state as well. It would be nice if the authors could separate these effects. Maybe some complement experiments with wt and inactive mutant would help. Or to show no chromatin RNA methylation involved in these loop formation.

Response: We thank the reviewer for the suggestion. Based on the reviewer's suggestions, we analyzed chromosome-associated regulatory RNAs (carRNAs) in control and senescent cells with or without METTL3 and METTL14 knockdown that we published previously. We showed that HK2 was not subject to the regulation of chromatin RNA methylation at putative distal enhancers. We have now included this result in the revised manuscript (New data in **Extended Data Fig. 2d**).

The authors could knock down YTHDC1, a nuclear reader of m6A, and presumably HK2 expression would not be affected.

Response: Based on the reviewer's suggestions, we knocked down YTHDC1 in senescent cells. As predicated by the reviewer, HK2 expression remained no significant change. We have now included this result in the revised manuscript (New data in **Extended Data Fig. 2j-k**).

HK2 might form stress granule and hijack YTHDF1 or YTHDF3, which are known to promote translation of methylated mRNA. I would suggest KD YTHDF1 or 3 and monitor translation efficiency change before HK2 activation. Overexpression of these genes may rescue the translation effect.

Response: We thank the reviewer for this insightful suggestion. As recommended, we first examined the HK2 expression at different time points in 4-OHT induced senescent model. Results showed that HK2 expression was not significantly induced at Day 1 (New data in **Extended Data Fig. 7h**). We then detect the translational efficiency of mRNAs including *CDCA8*, *CENPE*, and *SMC2* by polysome profiling through knockdown of YTHDF3 at Day 1. As predicated by the reviewer, knockdown of YTHDF3 did not increase the translational efficiency for transcripts such as *CDCA8*, *CENPE*, and *SMC2* prior to HK2 activation (New data in **Extended Data Fig. 7i-j**).

As a complementary approach, we conducted immunofluorescence staining for HK2 and stress granules and showed that there was no colocalization of HK2 with stress granules (New data in **Extended Data Fig. 6g**).

Reviewer #3 (Remarks to the Author):

In this study, the authors uncovered the functions of METTL3-mediated chromatin contacts in phase separation and metabolic reprogramming during senescence. The authors previously reported that m6A-independent genome-wide METTL3 and METTL14 redistribution drives the senescence-associated secretory phenotype (Nature Cell Biology 2021). Here they further showed that the MTC complex coordinates its enzymatic activity-dependent and -independent functions to regulate cellular senescence. In addition, METTL3-mediated chromatin loops induce Hexokinase 2 (HK2) expression through the three-dimensional chromatin organization during senescence. The elevated HK2 expression subsequently promotes liquid-liquid phase separation (LLPS) by driving metabolic reprogramming. Overall, this study provides the new mechanistic insights about the METTL3 enzymatic activity-dependent and -independent functions in regulating senescence. My specific comments are listed below:

Major:

1. Fig. 2a, given that the authors reported that METTL14 knockdown affect HK2, it's essential to address whether METTL14 also binds to HK2 promoter or enhancer?

Response: We thank the reviewer for this comment. To address whether METTL14 also binds to HK2 promoter or enhancer, we cross referenced H3K27ac ChIP-seq, METTL3 Cut&Run-seq, and METTL14 Cut&Run-seq along the *HK2* genomic region using datasets that we published previously [1]. The analysis revealed novel METTL14 bound peaks at putative distal enhancer sites in senescent compared to control cells, suggesting an increased binding sites of METTL14 at the enhancer regions of *HK2* (New data in **Figure 2a**). We have now included this result in the revised manuscript.

2. Though the authors identified a list of 38 genes that are regulated by MTC complex during senescence, it remains to be explained that why those genes are specifically regulated by MTC complex.

Response: We thank the reviewer for this insightful comment. Our study unbiasedly utilized HiChIP-seq and fastGro-seq to identify 38 genes that are regulated by MTC complex during senescence. Notably, these 38 genes enriched in crucial pathways associated with senescent alterations, including senescence-associated secretory phenotype (SASP) regulation, and metabolic reprogramming (**Figure 1e**). In a previous publication, we showed that SASP genes were transcriptionally regulated by MTC complex through its genome-wide redistribution, with NF- κ B subunit p65 contributing to the MTC-mediated regulation [1]. In the current study, we highlight the specific regulation of a metabolic gene *HK2* by the MTC complex through high-order chromatin organization (**Figure 2**). We further cross-referenced p65 Cut&Run-seq and METTL3 Cut&Run-seq along the *HK2* genomic locus (referred to #Review1-Question2). The analysis revealed an increased p65 bound peaks at putative distal enhancer loop anchors in senescent compared to control cells, indicating METTL3 may regulate high-order chromatin structure through its association with p65 (New data in **Extended Data Fig. 2c**).

3. The authors claimed that METTL3 promotes the transcription of HK2 independent of its catalytic activity by using the STM2457. In the original article (Reference 25), STM2457 was used to treat cells for a long time. Here the authors only treated the cells with low concentration for 48 hours, it remains to be validated that STM2457 did actually inhibit METTL3's enzymatic activity in the authors' experimental conditions.

Response: We agree with the reviewer. Our new results showed that the treatment condition used in the present study indeed significantly reduced m⁶A levels, demonstrating the inhibitory effect on METTL3 enzymatic activity by STM2457 treatment (New data in **Extended Data Fig. 2h**).

4. Similarly, to prove that METTL3 promotes the transcription of HK2 independent of its catalytic activity, more direct evidence is that the catalytic mutant of METTL3 can promote HK2 transcription.

Response: We thank for the reviewer for this comment. Based on the reviewer's suggestions, we checked whether the catalytic mutant of METTL3 can rescue HK2 transcription using our previously published datasets [1] and found that HK2 expression was decreased upon METTL3 knockdown, which was rescued by the catalytic mutant of METTL3 (New data in **Extended Data Fig. 2f**).

5. The authors concluded that METTL3 promotes HK2, which regulates purine metabolism. It remains to be demonstrated that whether METTL3 regulates purine metabolism, and whether METTL3's function in purine metabolism depends on HK2.

Response: We thank the reviewer's comments. Given that ATP and inosine are essential metabolites in purine metabolism, we measured their level following METTL3 knockdown and subsequent HK2 rescuing in senescent cells. Our finding revealed a reduction in ATP and inosine levels upon METTL3 knockdown, which was restored by HK2 expression, suggesting that METTL3's function in purine metabolism depends on HK2 (New data in **Extended Data Fig. 4c-e**).

Minor :

1. "HK2 is upregulated in senescent compared with control cells and the observed upregulation was rescued by knockdown of METTL3 or METTL14 (Extended Data Fig. 1b)." Extended Data Fig. 1b is not the data the authors mentioned.

Response: We apologize for this error, and we correct this accordingly.

2. NaAsO2, the 2 should be lower case?

Response: We have revised this typo accordingly.

References

1. Liu, P., F. Li, J. Lin, T. Fukumoto, T. Nacarelli, X. Hao, A.V. Kossenkov, M.C. Simon, and R. Zhang, *m(6)A-independent genome-wide METTL3 and METTL14 redistribution drives the senescence-associated secretory phenotype*. Nat Cell Biol, 2021. **23**(4): p. 355-365.

REVIEWERS' COMMENTS

Reviewer #1 (Remarks to the Author):

Authors have adequately addressed the raised comments and thus I would like to recommend the publication of this work.

Reviewer #2 (Remarks to the Author):

The authors have addressed all my comments. This is a very nice work.

Reviewer #3 (Remarks to the Author):

The authors have addressed my comments. The manuscript is ready for publication.

Point by point response to the reviewer's comments

REVIEWERS' COMMENTS

Reviewer #1 (Remarks to the Author):

Authors have adequately addressed the raised comments and thus I would like to recommend the publication of this work.

Response: Thank you.

Reviewer #2 (Remarks to the Author):

The authors have addressed all my comments. This is a very nice work.

Response: Thank you.

Reviewer #3 (Remarks to the Author):

The authors have addressed my comments. The manuscript is ready for publication.

Response: Thank you.